# EFFICIENT IMPLICIT NEURAL SURFACES VIA MULTI-SCALE RESIDUALS AND NESTED TRAINING

## ABSTRACT

Encoding input coordinates with sinusoidal functions into multi-layer perceptrons (MLPs) has proven effective for implicit neural representations (INRs) of surfaces defined as zero-level sets. This approach enables the capture of high-frequency detail and supports geometric regularization through MLP derivatives, such as the Eikonal constraint for signed distance function (SDF) fitting. However, existing methods typically rely on a single large MLP to learn the surface across the entire domain — a design that hinders efficient modeling of fine-grained details. Scaling the model may enable enhanced surface modeling, but at the cost of a larger number of MLP parameters and expensive inference, since mesh extraction or sphere tracing requires querying the MLP at many off-surface points. To address these issues, we propose M-plicits (**M**ultiscale Im**plicit** Neural **s**urfaces), a multiscale framework for representing and training INRs to encode surfaces as SDFs, enabling both high-quality reconstruction and efficient inference. To increase representational capacity, we model the INR as a residual sum of MLPs, where each component captures a specific level of detail, modulated by the sinusoidal input encodings. To improve efficiency, a small MLP captures coarse geometry, while finer residual MLPs are trained within a sequence of nested neighborhoods around the zero-level set. This design concentrates modeling capacity near the surface, improving reconstruction and reducing computation by relying on coarse approximations for off-surface points. Experiments show that M-plicits achieves state-of-the-art accuracy in surface reconstruction across standard benchmark datasets, while maintaining a compact representation. Our method also supports real-time sphere tracing and efficient high-resolution mesh extraction. Code and models will be released.

## 1 INTRODUCTION

Reconstructing surfaces from point clouds is a long-standing problem in vision and graphics (Kazhdan et al., 2006), with applications in augmented/virtual reality (Tkach et al., 2016), digital twins (Sun et al., 2005), cultural heritage preservation (Scopigno et al., 2011), and autonomous robotics (Whelan et al., 2016)—where high-quality 3D geometry is essential for perception, interaction, and decision-making. The emergence of high-resolution depth sensors has further motivated the development of accurate and efficient surface reconstruction methods. Representing the surfaces as zero-level sets of multi-layer perceptrons (MLPs) has became a prominent approach, due to their strong representational capacity and the ability to incorporate geometric regularizations (Wang et al., 2021). To enhance model expressiveness, input coordinates are projected into a set of sinusoidal functions (Tancik et al., 2020; Novello et al., 2025), allowing the network's bandlimit to be controlled through appropriate frequency initialization. For geometric regularization, it is common to assume that the underlying function represents a signed distance function (SDF) of the ground-truth surface (Schirmer et al., 2024), which satisfies the Eikonal equation. Incorporating this constraint into the implicit neural representation (INR) loss function serves as a geometric regularizer, helping to prevent overfitting.

Most existing INRs rely on a single large MLP to model the SDF across the entire domain. While scaling up the network can capture finer geometric details through high-frequency components, it also leads to large models with prohibitive inference costs, making them impractical for fast or real-time applications. This is especially limiting for level set extraction methods—such as marching cubes (Lorensen & Cline, 1987) or sphere tracing (Hart et al., 1989)—which require dense off-surface evaluations. For instance, SIREN (Sitzmann et al., 2020) requires full MLP evaluation at every

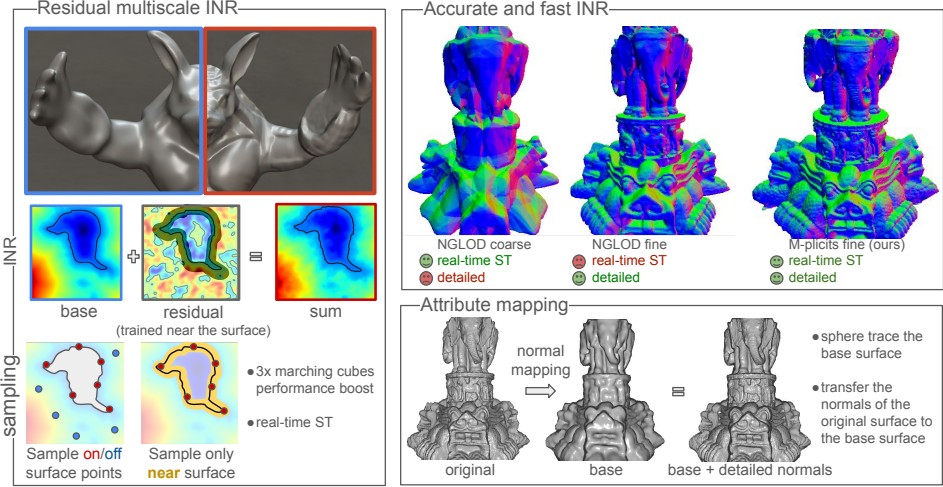

Figure 1: We introduce **M-plicits**, a multiscale INR framework for SDFs, based on a nested neighborhood scheme. (Left) Geometry is decomposed into base and residual SIRENs, trained at progressively finer scales with localized, near-surface sampling. (Top-right) M-plicits supports real-time sphere tracing (ST) with high fidelity, outperforming NGLOD (Takikawa et al., 2021) in both quality and rendering speed. (Bottom-right) M-plicits supports downstream tasks such as normal mapping.

query point, regardless of proximity to the surface, leading to substantial computational overhead. To alleviate this, grid-based MLPs have been proposed (Müller et al., 2022), enabling localized evaluations. However, introducing grid dependencies may limit surface smoothness and remain unsuitable for real-time rendering with high geometric fidelity.

To address these limitations, we introduce **M-plicits** (**M**ultiscale Im**plicit** Neural **S**urfaces), an INR framework for encoding SDFs in multiscale for efficient training and inference. M-plicits models the SDF of a surface as a residual sum of MLPs, where each MLP captures a different level of detail modulated by sinusoidal input encodings. A small, coarse MLP gives the global shape across the whole domain, while finer residual MLPs are trained within a sequence of nested neighborhoods around the previous zero-level sets. This multiscale strategy concentrates modeling capacity near the zero-level set, enhancing surface reconstruction accuracy and computational efficiency. In addition to geometry, M-plicits also models surface attributes—such as normals and textures—within the same neighborhood structure, ensuring smooth and consistent outputs. For real-time rendering, we introduce *multiscale sphere tracing* and a *normal computation* based on the General Matrix Multiplication (GEMM) (Dongarra et al., 1990) that leverages efficient matrix operations on the GPU. M-plicits supports fast surface extraction and integrates naturally into point clouds pipelines. Figure 1 showcases M-plicits's results on fitting and rendering SDFs, demonstrating significant improvements over prior methods in accuracy and inference speed. In summary, our contributions are:

- A compact and efficient multiscale INR model for accurate SDF representation, formulated as a residual sum of MLPs, each capturing a specific frequency band—achieving high representational capacity with fast inference for detailed surface modeling.

- A nested neighborhood training strategy that refines each residual component by supervising only near the previous zero-level set, enhancing geometric fidelity and improving data efficiency for oriented point clouds.

- M-plicits enables fast inference through real-time multiscale sphere tracing, a GEMM-based technique for normal computation, and efficient mesh extraction with support for normal and texture mapping, delivering SoTA performance in surface rendering and extraction.

## 2 RELATED WORK

Implicit representations are central to graphics and vision (Velho et al., 2007; Macêdo et al., 2009; Mescheder et al., 2019), with SDFs serving as a fundamental tool for modeling and manipulating surfaces (Bloomenthal & Wyvill, 1990; Sang et al., 2025). Recently, MLPs have been shown effective as INRs to model SDFs (Park et al., 2019; Gropp et al., 2020), including SIRENs (Sitzmann et al., 2020), which use periodic activations to capture high-frequency details.

**Multiscale Neural SDFs**. Various methods have investigated multiscale or frequency-aware training to enhance the expressiveness of INRs. BACON (Lindell et al., 2021) employs multiplicative filter networks (MFNs) (Fathony et al., 2020) to band-limit the spectrum of INRs. However, this approach introduces artifacts due to hard spectral truncation, and the lack of non-linear activations limits the capacity to represent fine details with small networks. Dou et al. (2023) improves the use of MFNs by integrating a feature grid and small architectural changes. BANF (Shabanov et al., 2024), MINER Saragadam et al. (2022), and MRNet (Paz et al., 2023) follow a multiscale design using a Laplacian pyramid. BANF uses grid-based MLPs, which increases memory usage and introduces a dependency on spatial grids. Also, these methods supervise off-surface regions at all scales, limiting their efficiency and making real-time inference more challenging. In contrast, our approach constructs a residual sum of SIRENs, with each component supervised to capture a distinct frequency band improving geometric fidelity and data efficiency by concentrating learning near the surface.

**Inference and Rendering**. Traditional visualization of SDFs relies on marching cubes (Lorensen & Cline, 1987) or sphere tracing (ST) (Hart, 1996). Performance-focused approaches such as (Davies et al., 2020) leverage ST to enable real-time rendering of INR level sets. NGLOD, in particular, interpolates hierarchical features from a sparse voxel octree, which are decoded by shallow MLPs. However, this interpolation results in discontinuous gradients, which impair normal estimation. Also, it cannot render highly-detailed level sets in real time. M-plicits addresses these issues by representing each frequency band explicitly using SIRENs, enabling smooth and efficient inference.

**Attribute Mapping**. Finally, classical attribute mapping techniques, such as normal mapping (Cohen et al., 1998), enhance surface detail but require explicit parameterizations and are sensitive to geometric distortions. Recent neural approaches (Wang et al., 2022) extend these ideas by propagating learned features off-surface using convolutional modules. Our method simplifies this process by avoiding interpolation entirely: inspired by variational inpainting techniques (Bertalmío et al., 2001), we regularize attribute fields to remain smooth along normals near the surface. For texture mapping, traditional parameterized approaches (Catmull, 1974) and neural texture fields (Oechsle et al., 2019; Gao et al., 2022) require known meshes or image-depth pairs and often involve complex training. In contrast, we define texture fields directly from colored point clouds using compact MLPs, regularized along the surface, enabling fast, texture-aware rendering without dense supervision or UV mapping.

## 3 M-PLICITS

Our goal is to model SDFs using a multiscale INR based on a residual sum of SIRENs, where each component captures a distinct level of detail. We train these networks using a *nested neighborhood* strategy: each residual is supervised only near the current zero-level set, concentrating learning near the surface. This improves geometric details and enables fast inference. Additionally, we introduce an attribute mapping scheme that leverages the SDF structure to support textures and normals without relying on mesh parameterizations or interpolation. Figure 2 gives an overview of our method.

### 3.1 PRELIMINARIES

Given an oriented point cloud $\{x_j, N_j\}_{j=1}^n$, consisting of surface points $x_j$ and their normals $N_j$, our goal is to reconstruct the underlying surface $S$ as the zero level set of a signed distance function (SDF) $f : \mathbb{R}^3 \to \mathbb{R}$, i.e., $S = f^{-1}(0) = \{x \mid f(x) = 0\}$, with the additional condition that $f(x_j) \approx 0$ and $\nabla f(x_j) \approx N_j$. To regularize the solution away from the input data, we also recall that true SDFs satisfy the *Eikonal equation*: $\|\nabla f(x)\| = 1$ for all $x \in \Omega$ in the training domain $\Omega$. Enforcing this condition during training improves generalization in unsupervised regions. Finally, combining the data constraints with the Eikonal regularization leads to a loss function (Gropp et al., 2020):

$$\mathcal{L}(f) = \frac{1}{n} \sum_j \left[ f(x_j)^2 + (1 - \langle \nabla f(x_j), N_j \rangle) \right] + \int_\Omega (\|\nabla f(x)\| - 1)^2 \, dx. \tag{1}$$

The first two terms ensure that the network fits the input points and aligns the gradient with the ground-truth normals. The third term enforces the Eikonal constraint, providing geometric regularization over $\Omega$. To capture fine geometric detail, it is common to parameterize the SDF $f$ using a sinusoidal MLP (SIREN). A SIREN with $n-1$ hidden layers is defined as:

$$f(x) = W_n \circ h_{n-1} \circ \cdots \circ h_0(x) + b_n, \quad \text{where } h_i(x) = \sin\left(\omega_0(W_i x + b_i)\right). \tag{2}$$

Here, $\omega_0$ is a frequency parameter controlling the network's capacity to model high-frequency details (Sitzmann et al., 2020), and each $W_i, b_i$ are the learnable weight matrices and biases.

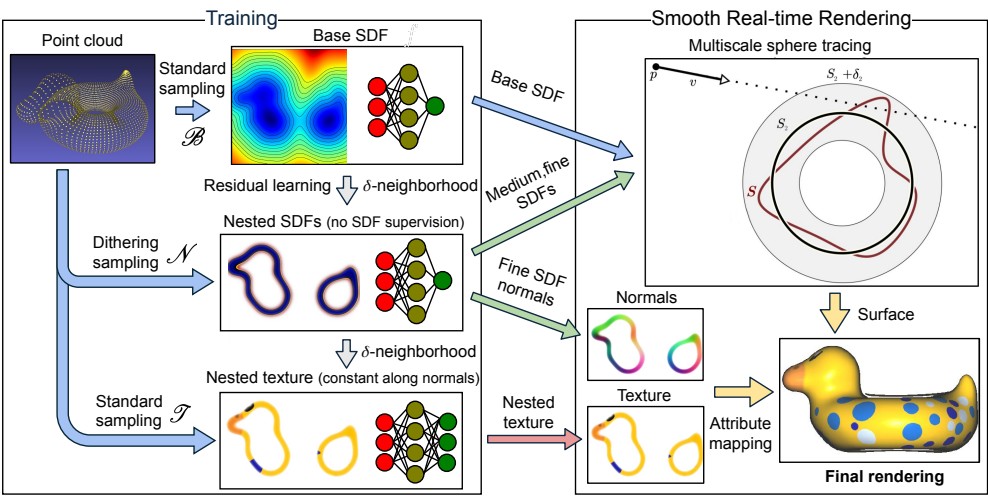

Figure 2: Overview of M-plicits. Starting from an oriented point cloud with colors, we combine sampling techniques and loss regularizations ($\mathscr{B}$ and $\mathscr{N}$) to create a base, a medium , and a fine SDF to implicitly represent the SDF in multiscale. The base SDF is defined for the entire domain, while the others are residuals, defined in (nested) neighborhoods of the surface. The colors are also trained in a neighborhood, regularized (by $\mathscr{T}$) to be constant along normals. The resulting multiscale representation can be rendered using novel sphere tracing and attribute mapping algorithms.

This architecture enables the network to fit both surface constraints $f(x_j) \approx 0$ and normal alignment $\nabla f(x_j) \approx N_j$, while providing high representational capacity for complex geometries. However, relying on a single large MLP to represent the entire domain is inefficient, especially for high-resolution surfaces, as it leads to high computational cost during inference.

## 3.2 MULTISCALE MODELING OF NEURAL SDFS

Standard neural SDFs typically rely on large MLPs to capture fine surface details, leading to fitting inefficiencies and high inference costs. To address this, we propose a multiscale SDF representation by recursively refining a coarse base network. Specifically, we define a sequence of neural SDFs $\{f_i\}_{i=1}^n$ via a sum of residual MLPs: $f_{i+1} = f_i + r_i$, for $i = 1, 2$, where $f_1$ is a compact base MLP trained over the full domain, and each residual $r_i$ refines $f_i$ by capturing higher-frequency components, modulated by the SIREN frequency parameter $\omega_0$. We refer to the sequence $\{f_i\}$ as *multiscale SDFs*, each approximating the SDF of $S$ at an increasing level of detail. While this formulation naturally extends to an arbitrary number of residuals, we focus on the case of three MLPs for simplicity and because this configuration performs well in our experiments.

To enable efficient learning and real-time inference, we impose a *nesting condition* between successive SDF levels. Specifically, each refined surface $S_i = f_i^{-1}(0)$, for $i = 2, 3$ is constrained to lie within a narrow band around the previous level set $S_{i-1}$, that is,

$$S_3 \subset \big[|f_2| < \delta_2\big] \subset \big[|f_1| < \delta_1\big] \tag{3}$$

This is enforced during training by supervising the residual MLP $r_i$ only within this $\delta_i$-neighborhood, a strategy we call *nested neighborhood training*. This condition ensures that each residual captures localized corrections, promotes coarse-to-fine refinement, and supports efficient applications such as progressive sphere tracing and surface-aware attribute mapping (see Section C). The choice of $\delta_i$ is tied to the training regime and is discussed in Section 3.2.

**Training with nested neighborhoods.** In standard SIREN-based SDF fitting, the loss in Equation 1 is applied across the entire domain $\Omega$. In contrast, our multiscale framework leverages the nesting condition to localize supervision. The hierarchy begins with a coarse-level SDF $f_1$, modeled by a compact SIREN. Finer levels $f_2, f_3, \ldots$ are added as residual SIRENs with progressively higher-frequency capacity, controlled by increasing values of the sinusoidal parameter $\omega_0$.

Training proceeds in stages: first, the base SDF $f_1$ is trained over the full domain $\Omega_0 := \Omega$; then, each subsequent level $f_{i+1}$ (for $i = 1, 2$) is trained within the restricted band $\Omega_i := [|f_i| < \delta_i]$. Each

SDF $f_i$ is optimized using a combination of data and Eikonal losses:

$$\mathcal{L}_i(f_i) = \frac{1}{n} \sum_j \left[ f_i(x_j)^2 + (1 - \langle \nabla f_i(x_j), N_j \rangle) \right] + \int_{\Omega_{i-1}} (\|\nabla f_i(x)\| - 1)^2 \, dx. \tag{4}$$

To ensure that $\{x_j\}$ lies within $\Omega_{i-1}$ at each stage $i$, we set the band threshold $\delta_i$ adaptively following:

$$\delta_i = (1 + \varepsilon) \cdot \max_j |f_i(x_j)|, \text{ with } \varepsilon > 0 \text{ a small threshold.} \tag{5}$$

**Sampling**. To discretize the Eikonal term over $\Omega_{i-1}$, we employ dithering-based sampling around the input points $\{x_j\}$, perturbing each coordinate by a random value in the interval $(-2\delta_{i-1}, 2\delta_{i-1})$. Samples falling outside $\Omega_{i-1}$ are rejected using the condition $|f_{i-1}(x)| < \delta_{i-1}$; see Fig. 3(a).

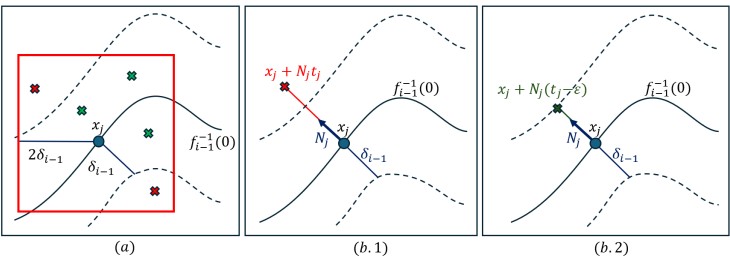

Figure 3: **(a)** Dithering-based sampling around each input point $x_j$, where each coordinate is perturbed by a random value in $(-2\delta_{i-1}, 2\delta_{i-1})$, followed by filtering to retain only points in the valid band $\Omega_{i-1}$ (green points denote those that are kept and red those rejected). **(b.1)** Computation of the displacement vector $t_j N_j$, where the offset point $x_j + t_j N_j$ lies just outside the narrow band $\Omega_{i-1}$. **(b.2)** Final accepted sample $x_j + N_j(t_j - \varepsilon)$, located inside $\Omega_{i-1}$, used to supervise the data term.

Adding extra samples to accelerate SDF training is a common strategy (Novello et al., 2022), but it often requires evaluating a large MLP over the entire domain, leading to inefficiencies and costly inference. In contrast, M-plicits uses residual MLPs that are trained only within a narrow band around the input data, allowing for efficient sampling. To enrich supervision of the data term in Equation 4, we propose sampling along the normal $N_j$ of each point $x_j$, as shown in Fig. 3(b). Specifically, we compute a scalar $t_j \leq \delta_{i-1}$ such that for all $t \in [0, t_j]$, the offset point $x_j + tN_j$ lies at a distance $t$ from the

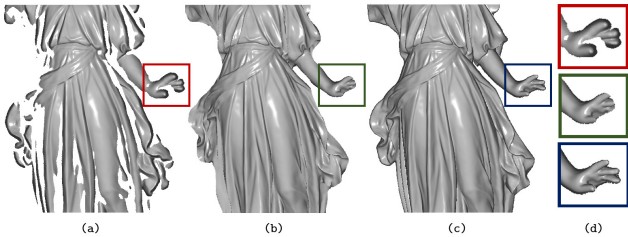

Figure 4: (a) Using too large $\delta_1 = \delta_2$ may result in holes (the ray does not reach the surface). More iterations would be needed using the finer SDF to fill those holes, defeating the idea of minimizing iterations. (b) Conversely, reducing the deltas $\delta_1 = \delta_2$ may miss parts of the silhouette since the target surface may not be inside the previous neighborhood (the hand). (c) Using $\delta_1$ and $\delta_2$ suited for the nesting condition implies in no holes and a better silhouette capture.

surface. This allows us to supervise both the SDF value and its gradient: $f_i(x_j + tN_j) = t$ and $\nabla f_i(x_j + tN_j) = N_j$, providing richer sampling near the surface. We determine each $t_j$ via a simple iterative scheme: starting from $t_j = \delta_{i-1}$, we reduce it by a small step until the distance from $x_j + t_j N_j$ to the point cloud $\{x_j\}$ equals $t_j$, ensuring that the offset point lies inside the band.

**Rendering and mesh extraction** To render the zero level set $f^{-1}(0)$ of an SDF $f$, it is common to use either sphere tracing (ST) or marching cubes for mesh extraction followed by standard mesh rendering. When the SDF is represented using M-plicits, both strategies become more efficient.

We first introduce a **multiscale ST**. Given a view ray $\gamma(t) = p_0 + tv$, with origin at a point $p_0$ and direction $v$, intersecting $f^{-1}(0)$, standard ST approximates the first intersection point by iterating $p_{i+1} = p_i + v f(p_i)$ along $\gamma$. However, querying a large MLP at each step may be expensive. To reduce this cost, we exploit the multiscale SDF hierarchy $\{f_i\}$, using coarser networks to guide early steps. Thanks to the nesting condition in Equation 3, coarse levels can be used to trace offset surfaces before switching to finer levels near the surface. The ray starts tracing $f_1^{-1}(\delta_1)$ with $f_1$, then proceeds to

$f_2^{-1}(\delta_2)$ using $f_2$, and finally reaches the target surface $f_3^{-1}(0)$ with $f_3$. Each coarser level uses offset tracing via $p_{i+1} = p_i + v\left(f_j(p_i) - \delta_j\right)$, ensuring convergence avoiding high-cost evaluations. The values $\delta_i$ (Equation 5) play a crucial role in rendering. Using distinct values at each level helps prevent issues such as missed ray-surface intersections: if $\delta_1$ is too small, parts of $f_2^{-1}(0)$ might lie outside the region bounded by $f_1^{-1}(\delta_1)$; Fig. 4 shows some of these issues.

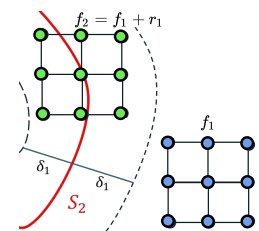

M-plicits also accelerates mesh extraction using **marching cubes**. We adopt an adaptive grid inference strategy by first evaluating the coarse SDF $f_1$ to cull grid vertices, and querying finer SDFs only for vertices inside the $\delta_1$-neighborhood. This reduces the number of voxel evaluations, thereby accelerating mesh extraction (Fig. 5).

**Normal and texture mapping.** Let $S$ be a surface nested within a $\delta$-neighborhood of the zero-level set of a neural SDF $f$. Assuming $f$ is a finer-level neural SDF, we define the *neural normal mapping* by assigning to each point $p \in S$ the attribute $g(p) := \nabla f(p)$. If $S$ corresponds to the zero-level set of a coarser neural SDF, this neural normal mapping allows us to bypass additional sphere tracing (ST) iterations, reducing computational overhead.

Figure 5: Adaptive marching cubes. For grid vertices outside the $\delta_1$ neighborhood (blue), only the coarse SDF $f_1$ is evaluated. The residual $f_2$ is added for other points (green).

Similarly, we define a network $g \colon \mathbb{R}^3 \to \mathcal{C}$ to encode a *texture* within the $\delta$-neighborhood of $f$, where $\mathcal{C}$ is typically the RGB color space. We refer to the attribute mapping defined by the triple $\{S, f, g\}$ as a *neural texture mapping*. To train the parameters $\phi$ of $g$, we optimize $\mathcal{T}(\phi) = \int_{f^{-1}(0)} (g - g)^2\, dx + \int_{[|f| \le \delta]} \langle \nabla g, \nabla f \rangle^2\, dx$, where the first term ensures that $g$ fits the ground-truth texture $g$, while the second term regularizes $g$ to remain constant along the gradient flow of $f$, effectively propagating texture information throughout the $\delta$-neighborhood.

**GEMM-based normal calculation.** To ensure real-time rendering performance, we compute normals without the need of auto-differentiation nor computational graphs. It works as a forward pass on the MLP and is implemented on the GPU using only a GEMM library, resulting in a 2X performance improvement over `torch.autograd`. Implementation details are in Section E.

## 4 Experiments

This section presents experiments to evaluate the proposed method comprehensively, both in comparison to the state of the art and with ablations designed to understand the importance of its different components. We also demonstrate several applications of our method, showcasing its versatility.

**Implementation details.** All experiments are conducted on an NVidia RTX 5090. For sphere tracing, we fix the number of iterations to 20 for coarse and 5 for each residual level, for better control of the parallelism. The $\omega_0$ SIREN parameters are 40 for coarse, 80 for medium, and 100, 128 or 180 for fine, depending on complexity. We use PyTorch's Adam optimizer for training (Paszke et al., 2019).

**Evaluation protocol.** All input point clouds were centered and normalized to the unit sphere. For evaluation, we extracted the zero-level set of each Neural SDF using marching cubes at a resolution of $512^3$, followed by re-normalization for consistency. We then uniformly sampled 500K points on the reconstructed meshes and computed the L2 Chamfer Distance (CD) against the input point cloud using PyTorch3D's implementation (Ravi et al., 2020). Importantly, although widely adopted in SDF benchmarks, CD values are not directly comparable across papers, as they depend on choices such as normalization (sphere vs. box), metric (L1 vs. L2), and sampling strategy. Establishing a robust evaluation protocol remains challenging in the field and outside the scope of this work.

**MLP notation.** $(N, d)$ refers to a MLP with $d$ hidden layers of the form $\mathbb{R}^N \to \mathbb{R}^N$. Additionally, $(64, 2) \triangleright (128, 2) \triangleright (256, 2)$ refers to a multiscale SDF with coarse, medium, and fine MLPs with two $\mathbb{R}^{64} \to \mathbb{R}^{64}$, $\mathbb{R}^{128} \to \mathbb{R}^{128}$, and $\mathbb{R}^{256} \to \mathbb{R}^{256}$ hidden layers, respectively.

### 4.1 Main results

**Surface reconstruction.** We compare our neighborhood nesting approach against SoTA methods for surface representation on the Stanford dataset (Curless & Levoy, 1996). We compare with NGLOD and Instant-NGP (Müller et al., 2022), focusing on real-time rendering performance, as well as

Table 1: **Surface reconstruction: comparison to the state of the art.**. Best values are **bold**, second best are underlined, and third best are *italic*. Chamfer Distance (CD) considers 500K samples. Sampling time is measured on a $512^3$ grid. Real-time renderer throughput considers $512^2$ images. M-plicits has the best CD metrics, with competitive parameter count and training time. Its runtime performance is highlighted by the best sampling times and real-time renderer throughput. NGLOD has a real-time setup that considerably compromises surface quality, as shown by the CD column.

| Input | Method | CD ↓ | # params ↓ | Training (min) ↓ | Sampling (s) ↓ | Renderer FPS ↑ |
|---|---|---|---|---|---|---|
| Thai statue | iNGP fine | 8.60E-03 | 9,113,760 | **0** | N/A | 103 |
|  | iNGP coarse | 8.86E-03 | 2,040,864 | **0** | N/A | *100* |
|  | NGLOD coarse | 3.84E-03 | **8,737** | 162.68 | *3.81* | 40 |
|  | NGLOD fine | *3.52E-03* | 10,146,213 | 179.18 | 6.82 | N/A |
|  | IDF | **5.18E-04** | 1,191,943 | *20.57* | 28.10 | N/A |
|  | BACON | 1.95E-03 | 530,953 | 84.00 | 8.00 | N/A |
|  | Ours coarse | 4.36E-03 | 17,153 | 11.73 | **1.25** | 315 |
|  | Ours fine | 4.14E-03 | *132,865* | 39.68 | 1.73 | 70 |
| Asian Dragon | iNGP fine | 1.46E-02 | 9,113,760 | **0** | N/A | 87 |
|  | iNGP coarse | 1.47E-02 | 2,040,864 | **0** | N/A | *83* |
|  | NGLOD coarse | 7.25E-03 | **8,737** | 113.83 | *3.70* | 40 |
|  | NGLOD fine | 6.91E-03 | 10,146,213 | 127.28 | 6.80 | N/A |
|  | IDF | 4.40E-04 | 1,191,943 | 21.03 | 28.60 | N/A |
|  | BACON | 2.97E-05 | 530,953 | 38.10 | 8.30 | N/A |
|  | Ours coarse | *3.05E-05* | 17,153 | 8.60 | **1.25** | 315 |
|  | Ours fine | **1.03E-05** | *132,865* | 28.70 | 1.71 | 70 |
| Lucy | iNGP fine | 9.50E-03 | 9,113,760 | **0** | N/A | *92* |
|  | iNGP coarse | 9.46E-03 | 2,040,864 | **0** | N/A | 98 |
|  | NGLOD coarse | 3.44E-03 | 8,737 | 44.70 | *3.57* | 40 |
|  | NGLOD fine | 3.27E-03 | 10,146,213 | 61.13 | 6.58 | N/A |
|  | IDF | **2.46E-06** | 1,191,943 | *9.06* | 66.87 | N/A |
|  | BACON | 6.07E-04 | 530,953 | 165.33 | 7.77 | N/A |
|  | Ours coarse | *3.28E-04* | **4,481** | 1.81 | **0.69** | 315 |
|  | Ours fine | 6.83E-05 | *162,401* | 10.36 | 1.07 | 70 |
| Armadillo | iNGP fine | 1.93E-02 | 9,113,760 | **0** | N/A | 71 |
|  | iNGP coarse | 1.88E-02 | 2,040,864 | **0** | N/A | *63* |
|  | NGLOD coarse | 1.44E-02 | 8,737 | 14.20 | *3.80* | 40 |
|  | NGLOD fine | 1.43E-02 | 10,146,213 | 31.58 | 6.85 | N/A |
|  | IDF | 6.55E-04 | 1,191,943 | 2.23 | 37.68 | N/A |
|  | BACON | **8.27E-05** | 530,953 | 40.88 | 8.28 | N/A |
|  | Ours coarse | 1.38E-04 | **2,593** | *4.97* | **0.50** | 315 |
|  | Ours fine | *2.06E-04* | *67,073* | 5.42 | 0.99 | 70 |

with Implicit Displacement Fields (IDF)(Wang et al., 2022), which disentangles shape and detail to capture fine surface structure. Finally, we include BACON (Lindell et al., 2021), a well-established multiscale approach, where we report results from its 8th hidden layer. Tab. 1 summarizes the results. Despite enabling real-time rendering and achieving the fastest sampling times, our method shows training times comparable to IDF, which relies on mesh extraction for rendering. Training is up to one order of magnitude faster than NGLOD at similar surface detail, and up to $4.5\times$ faster in fine-scale reconstructions. Importantly, our approach does not sacrifice surface quality for efficiency: it consistently achieves the best, second-best, or third-best Chamfer Distance (CD) across all cases. Fig. 6 shows render comparisons using the Armadillo. NGLOD level 0 is able to render geometry in real time, however, it presents significantly lower details compared to the others. To avoid this, we use an NGLOD level 5 configuration, which has less discretization artifacts, albeit forgoing its real-time rendering capability. M-plicits achieves real-time rendering performance, while maintaining a detailed smooth surface, unlike NGLOD. Also, IDF requires mesh extraction, thus not being directly renderable in real-time. Instant-NGP offers extremely fast training and real-time rendering but at the cost of higher CD and parameter counts—up to two orders of magnitude larger than ours. For fairness, we compared against both a coarse Instant-NGP setup with 3 levels of detail and a fine setup with 16 levels. Note that none of these competitor methods support textures.

Although the coarse and fine cases have close CD for some surfaces, Figure 17 (cols. 1 and 3) shows that they are very different perceptually. Moreover, our approach is able to map normals from detailed surfaces to include detail faster (col. 2).

**Normals.** We compare our GEMM normal calculation against `torch.autograd`. Our method performs $2\times$ faster across 6 different INRs trained for the Armadillo, Buddha and Lucy, with architectures varying between 2 and 3 hidden layers. More details are provided in Tab. 10.

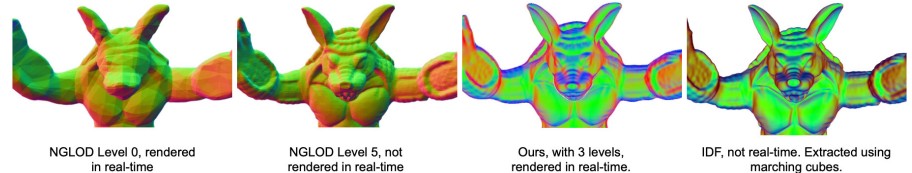

NGLOD Level 0, rendered
in real-time

NGLOD Level 5, not
rendered in real-time

Ours, with 3 levels,
rendered in real-time.

IDF, not real-time. Extracted using
marching cubes.

Figure 6: Armadillo renderings using different methods. From the left: NGLOD levels 0, 5, our method, and IDF. Note that NGLOD level 0 and our method are rendered in real time with sphere tracing. NGLOD level 5 is not real time. IDF was rendered using the rasterization pipeline after running marching cubes.

## 4.2 ABLATIONS AND ADDITIONAL EXPERIMENTS

**Residuals.** To evaluate our nested neighborhood scheme and loss design, we compare against a **baseline** residual variant $f_2 = f_1 + r_1$, where both $f_1$ and $r_1$ are trained using SIREN's original loss and sampling strategy, following the baseline in IDF (Wang et al., 2022). In contrast, our method employs the loss in Equation 4 together with *nested neighborhoods* to supervise the ground-truth SDF within a narrow band around the previous stage's zero level set. This exploits the property $f(\mathbf{x}_j + t\,\mathbf{N}_j) = t$ within a tubular neighborhood, enabling supervision beyond the original samples $\mathbf{x}_j$ to improve stability and prevent error growth. This strategy also reduces network complexity, allowing us to use much smaller architectures: a base network with a single hidden layer of 128 neurons ($\omega_0 = 30$) and $r_1$ with a single hidden layer of 256 neurons ($\omega_0 = 45$).

We compare this baseline with M-plicits on the Thingi32 dataset, where the baseline achieves an average CD of 6.2E-2, while our method reaches 1.2E-2, demonstrating that M-plicits yields substantially better reconstructions. We also evaluate the residual approach qualitatively. Fig. 8 shows that residuals eliminate spurious components when combined with neighborhood training. We further exploit this property to accelerate marching cubes in scenarios where mesh extraction is required.

**Neural normal mapping and multiscale ST.** Fig. 7 shows the case where the coarse surface is the zero-level of a neural SDF (left) and when it is a triangle mesh (middle), showing that our representation can also be beneficial for rendering meshes. An overall evaluation of the algorithm with other models is provided in the appendix. In all cases, normal mapping increases fidelity. The result may be improved using the multiscale ST, as shown in Fig. 7 (right). Adding ST iterations using a neural SDF with a better approximation of the surface improves the silhouette (right).

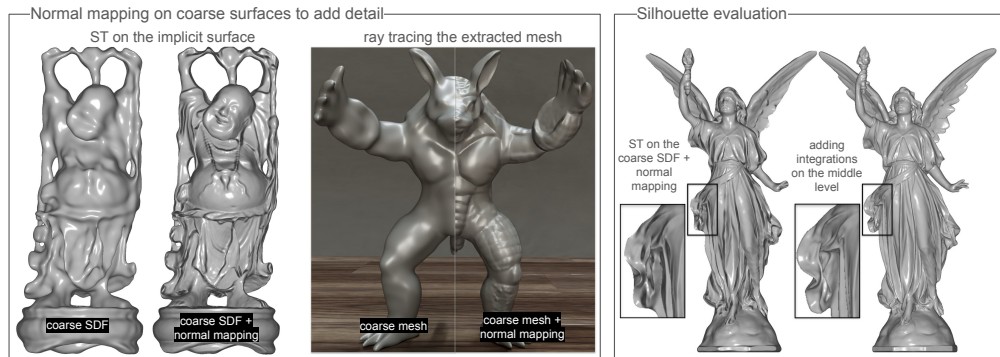

Figure 7: Left: neural normal mapping onto a neural SDF. First, the coarse $(64, 1)$ SDF. Then, the neural normal mapping of the $(256, 3)$ SDF onto the $(64, 1)$. Middle: neural normal mapping onto half of a triangle mesh. The normals of the $(256, 3)$ SDF are used. The mesh is the marching cubes of the $(64, 1)$ SDF. The *mean square error* (MSE) is $0.00262$ for the coarse case and $0.00087$ for the normal mapping, an improvement of $3\times$. The baseline is the marching cubes of the $(256, 3)$ SDF. Right: Silhouette evaluation. First a $(64, 1) \triangleright (256, 3)$, then a $(64, 1) \triangleright (256, 2) \triangleright (256, 3)$ configuration. Notice how the silhouette improves with the additional $(256, 2)$ level.

**Real-time renderer.** We evaluate a GPU version implemented in a CUDA renderer, using neural normal mapping, multiscale ST, and the GEMM-based analytical normal calculation (implemented using CUTLASS). Tab. 2 shows the results. Notice that the framework achieves real-time performance and that using neural normal mapping and multiscale ST improves performance considerably.

The flexibility of our multiscale SDF representation enables additional applications, including integration into differentiable pipelines and fast mesh extraction using the marching cubes algorithm.

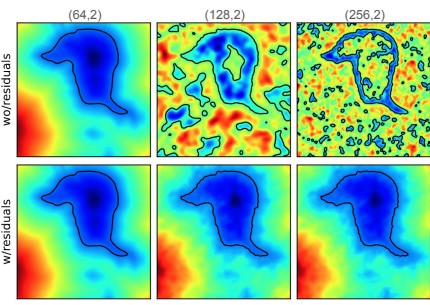

Figure 8: Evaluation of the residual approach. Note that training the SDFs in the neighborhoods (first row: center, right) results in spurious components outside the region as would be expected. Using the residual approach eliminates those components (second row: center, right).

Table 2: In-depth ablation of the real-time CUDA renderer using multiscale ST, GEMM normals, and normal mapping. The number of iterations is 20 for the first neural SDF and 5 for the subsequent ones. (NM) indicates normal mapping and no ST iterations for the last SDF. Images are $512^2$. Size is in KB. Note that the residual approach allows smaller networks and that all cases result in speedups. M-plicits are very small and flexible, being easily adaptable to different performance budgets.

| Model | FPS | Speedup | Size |
|---|---|---|---|
| $(256, 4)$ (SIREN baseline) | 37 | 1.0X | 777 |
| $(64, 2)$ (coarse) | 315 | 8.5X | 11 |
| $(64, 2) \triangleright (128, 2)$ (NM) | 177 | 4.1X | 79 |
| $(64, 2) \triangleright (128, 2)$ | 150 | 4.1X | 79 |
| $(64, 2) \triangleright (256, 2)$ (NM) | 94 | 2.5X | 274 |
| $(64, 2) \triangleright (256, 2)$ | 75 | 2.0X | 274 |
| $(64, 2) \triangleright (128, 2) \triangleright (256, 2)$ (NM) | 86 | 2.3X | 342 |
| $(64, 2) \triangleright (128, 2) \triangleright (256, 2)$ | 70 | 1.9X | 342 |

**Mesh extraction**. Experiments show that M-plicits improves the performance of grid evaluation by avoiding inference at finer levels far from the level set. Tab. 3 presents a maximum performance improvement of $5\times$, and surface reconstructions are given in Fig. 19. For all cases, the baseline SDF is approximated by a single MLP $(256, 4)$, while the multiscale SDFs have a configuration of $(64, 1) \triangleright (128, 2) \triangleright (256, 2)$. Note that surfaces occupying smaller domain regions have a greater speedup since the number of vertices in their nesting neighborhoods decreases.

Table 3: Marching cubes runtime comparison in seconds. The baseline is a SIREN network with 4 hidden layers with 256 neurons. Using our multiscale surface representation results in up to $\mathbf{5\times}$ speedup compared to baseline.

| Model | Baseline | No culling | Culling | Speedup |
|---|---|---|---|---|
| Arm. | 4.87 | 4.10 | **0.99** | **4.92×** |
| Lucy | 4.87 | 7.49 | **1.07** | **4.56×** |
| Dragon | 4.88 | 7.28 | **1.71** | **2.85×** |
| Thai | 4.89 | 7.28 | **1.73** | **2.82×** |

**Textures.** We define textures directly in a neighborhood of the surface, removing the need for a UV map. This formulation produces visually convincing appearance while decoupling texture from geometry in a compositional way. To assess accuracy, we compared our approach against traditional UV-textured meshes by measuring the MSE between rendered images. Across five test models—Spot, Bob, Bunny, Egg, and Earth—we obtained MSEs of 0.0329, 0.0434, 0.0720, 0.0291, and 0.0033, respectively. Figure 9 illustrates neural texture mapping applied to coarse surfaces.

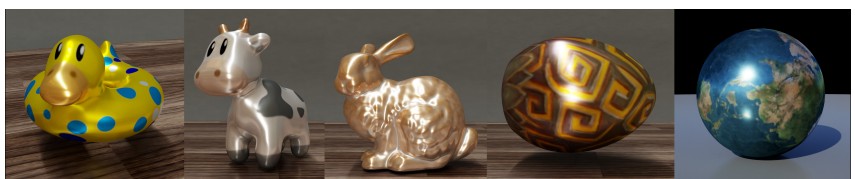

Figure 9: Neural texture mapping. All networks are $(256, 3)$, except for the the earth, which is $(512, 3)$. The surfaces are marching cubes of $(64, 1)$ SDFs, except for the bunny, which is $(128, 2)$.

**Robustness to noise.** To evaluate the robustness of M-plicits to noisy point clouds, we test it on a perturbed version of the *Lucy* model, where all vertices are randomly perturbed in the direction of the normal by at most 1.0% of the model bounding box. As shown in Figure 10, the coarse level of M-plicits acts as a low-pass filter, removing most of the high-frequency noise and providing a clean geometric prior that benefits the subsequent residual levels. In contrast, Instant-NGP struggles under this noise regime and fails to recover a smooth and coherent surface. We choose Instant-NGP because it is a strong baseline for real-time models.

**Scene scale test.** We also evaluated our method on a scene-scale point cloud (Figure 16). M-plicits successfully reconstructs the entire scene across all scales, whereas Instant-NGP fails in our tests. The Chamfer distance further corroborates these observations: M-plicits achieves 2–3 orders of magnitude lower Chamfer distance.

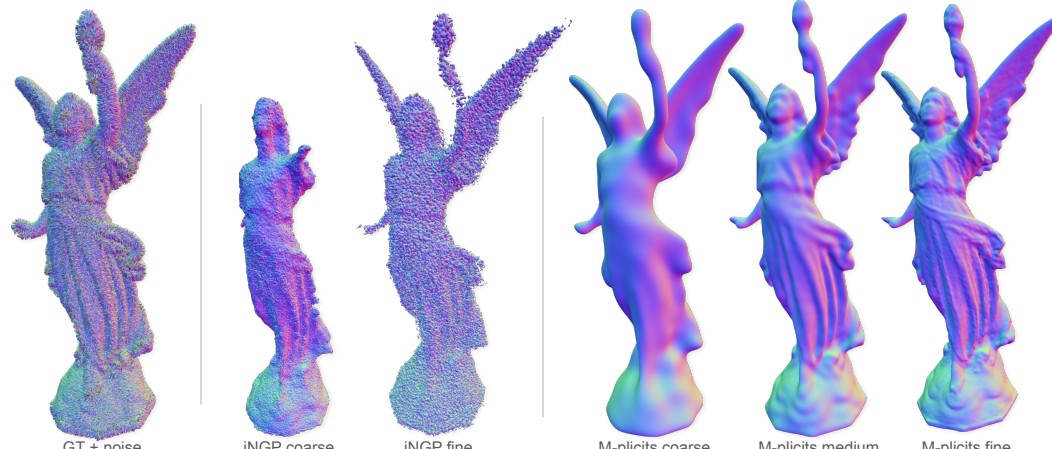

GT + noise      iNGP coarse      iNGP fine      M-plicits coarse      M-plicits medium      M-plicits fine

Figure 10: **Robustness to noise.** Comparison of reconstructions from a noisy point cloud of the *Lucy* model. Instant-NGP (coarse/fine) fails to denoise the input and produces highly irregular surfaces. In contrast, the coarse level of M-plicits removes most of the noise, while the medium and fine levels progressively refine the geometry.

Quantitative results for three models using Chamfer, Hausdorff, and IoU further confirm the robustness of our approach, as shown in Table 4.

Table 4: **Quantitative robustness to noise.** Chamfer distance ($\downarrow$), Hausdorff distance ($\downarrow$), and IoU ($\uparrow$) for three models under vertex noise perturbation. Best values are **bold**, second best are *italic*, and third best are underlined.

| Model | Chamfer ($\downarrow$) | Hausdorff ($\downarrow$) | IoU ($\uparrow$) |
|---|---|---|---|
| **Armadillo** | | | |
| Ours coarse | *1.83E-03* | *1.18E-02* | *5.68E-02* |
| Ours medium | **5.86E-05** | **7.18E-04** | **4.89E-01** |
| Ours fine | 2.37E-03 | 1.21E-02 | 5.09E-02 |
| iNGP coarse | 1.75E-02 | 1.05E-01 | 3.54E-02 |
| iNGP fine | 1.93E-02 | 9.49E-02 | 3.34E-02 |
| **Asian Dragon** | | | |
| Ours coarse | 3.49E-04 | 1.55E-02 | 2.97E-01 |
| Ours medium | *2.02E-04* | 4.10E-03 | *3.70E-01* |
| Ours fine | **1.69E-04** | **2.76E-03** | **4.00E-01** |
| iNGP coarse | 2.35E-02 | 2.21E-01 | 4.70E-03 |
| iNGP fine | 1.62E-02 | 1.52E-01 | 6.68E-03 |
| **Lucy** | | | |
| Ours coarse | 3.33E-04 | 2.21E-02 | 3.02E-01 |
| Ours medium | *4.39E-05* | *8.31E-04* | *5.18E-01* |
| Ours fine | **3.45E-05** | 8.54E-04 | **5.98E-01** |
| iNGP coarse | 2.36E-02 | 1.36E-01 | 0.00E+00 |
| iNGP fine | 1.12E-02 | 7.57E-02 | 1.00E-03 |

## 5    CONCLUSION

We propose an INR framework to render surfaces in real-time using neural SDFs endowed with smooth normals and textures. It uses spatial neighborhoods and residual training, achieving real-time performance without the need for spatial data structures. The multiscale sphere tracing accelerates surface evaluation, the neural attribute mapping transfers surface attributes between surfaces, and the GEMM-based normal computation gives smooth normals without the need of auto-grad. Moreover, we show that our multiscale SDF can be used to accelerate mesh extraction using marching cubes.

**Limitations and future work.** As is common for SDF-based representations, our approach is not suited to represent very sharp edges. This is a natural consequence of the function smoothness and may be solved by incorporating local features into the function, a path we would like to explore in future work. The multiscale ST could probably be applied into neural SDF-based 3D reconstruction or inverse rendering tasks to reduce the training time. Nested neighborhoods could be adapted for unsigned distance functions too. Improvements can be done for further performance optimization. For example, using fully fused GEMMs may decrease the overhead of GEMM setup (Müller, 2021).

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

## A ADDITIONAL COMPARISONS

**Quantitative results in homogenized setups.** We performed additional experiments on a subset of the Thingi32 dataset using a homogenized setup. For all meshes, we employed a coarse model with a single hidden layer of 128 neurons and $\omega_0 = 30$, a medium model with a single hidden layer of 256 neurons and $\omega_0 = 45$, and a fine model with a single hidden layer of 400 neurons and $\omega_0 = 128$. Each stage was trained for 1000 epochs, for a total of 3000 epochs per mesh. We compare against

Instant NGP (Müller et al., 2022). This setup allows for relatively fast training (5–10 minutes on an RTX 4090), with a memory footprint of 4–6 GB (batch sizes of 125k and 75k for the medium and fine levels, respectively). The results in terms of CD are shown in Table 5. Our method achieves better metrics across all meshes. It is important to note, however, that this dataset does not match the quality of the Stanford dataset, as it contains self-intersections, holes, and poorly triangulated regions.

Table 5: Comparison with Instant NGP using a homogenized setup on a subset of the Thingi32 dataset. The iNGP-coarse model has 3 detail levels and the iNGP-fine model has 16 levels.

| Mesh Id | M-plicits (ours) | iNGP-coarse | iNGP-fine |
|---|---|---|---|
| 47984 | **1.17E-03** | 3.83E-02 | 3.98E-02 |
| 68380 | **8.44E-03** | 2.58E-02 | 2.54E-02 |
| 354371 | **1.69E-03** | 2.08E-02 | 2.10E-02 |
| 398259 | **6.00E-04** | 1.22E-01 | 1.25E-01 |
| 527631 | **1.45E-03** | 2.74E-03 | 3.03E-03 |

## B  ABLATION STUDIES

We performed two additional ablation studies for our approach: (i) loss term assessment, (ii) $\delta$ influence over the reconstructions. Tables 6,7, and 8 show the ablation results of our loss function using different weights for each component, while maintaining the remaining hyper-parameters fixed. We performed these studies both for a single intermediate level (medium) and an additional refinement level beyond it (fine). Note that all studies used the Lucy mesh as a baseline. Table 9 shows the results for varying the delta values while maintaining the remaining hyper-parameters fixed.

Table 6: Gradient constraint ablation studies.

(a) Gradient constraint fine level

| Gradient constraint | Approx. Error |
|---|---|
| 0.0 | 0.0013 |
| 10.0 | 0.0013 |
| 30.0 | 0.0013 |
| 100.0 | 0.0012 |
| 300.0 | 0.0013 |
| 1000.0 | 0.0014 |
| 3000.0 | 0.0017 |
| 10000.0 | 0.0022 |
| 30000.0 | 0.0030 |

(b) Gradient constraint medium level

| Gradient Constraint | Approx. Error |
|---|---|
| 0.0 | 0.0086 |
| 10.0 | 0.0084 |
| 30.0 | 0.0082 |
| 100.0 | 0.0078 |
| 300.0 | 0.0074 |
| 1000.0 | 0.0069 |
| 3000.0 | 0.0073 |
| 10000.0 | 0.0087 |
| 30000.0 | 0.0116 |

Table 7: Normal constraint ablation studies.

(a) Normal constraint fine level.

| Normal Constraint | Approx. Error |
|---|---|
| 0.0 | 0.0017 |
| 10.0 | 0.0013 |
| 30.0 | 0.0013 |
| 100.0 | 0.0013 |
| 300.0 | 0.0013 |
| 1000.0 | 0.0013 |
| 3000.0 | 0.0013 |
| 10000.0 | 0.0013 |
| 30000.0 | 0.0013 |

(b) Normal constraint medium level.

| Normal Constraint | Approx. Error |
|---|---|
| 0.0 | 0.0073 |
| 10.0 | 0.0074 |
| 30.0 | 0.0077 |
| 100.0 | 0.0081 |
| 300.0 | 0.0083 |
| 1000.0 | 0.0085 |
| 3000.0 | 0.0086 |
| 10000.0 | 0.0087 |
| 30000.0 | 0.0087 |

**Normals:** We compare our GEMM normal calculation against `torch.autograd`. As shown in Tab. 10, ours performs 2× faster. We tested 6 different INRs trained for Armadillo, Buddha, and Lucy, varying between 2-3 hidden layers.

Table 8: SDF constraint ablation studies.

(a) SDF constraint fine level.

| SDF Constraint | Approx. Error |
|---:|---:|
| 0.0 | 0.0076 |
| 10.0 | 0.0013 |
| 30.0 | 0.0013 |
| 100.0 | 0.0013 |
| 300.0 | 0.0013 |
| 1000.0 | 0.0012 |
| 3000.0 | 0.0013 |
| 10000.0 | 0.0013 |
| 30000.0 | 0.0013 |

(b) SDF constraint medium level.

| SDF Constraint | Approx. Error |
|---:|---:|
| 0.0 | 0.0490 |
| 10.0 | 0.0080 |
| 30.0 | 0.0079 |
| 100.0 | 0.0079 |
| 300.0 | 0.0080 |
| 1000.0 | 0.0081 |
| 3000.0 | 0.0080 |
| 10000.0 | 0.0082 |
| 30000.0 | 0.0082 |

Table 9: Ablation studies of the delta factor. We multiply the delta by the values in the first column and measure the SDF error compared to the Open3D calculated SDF, which we use an ground-truth.

| Max delta fraction | Medium level error | Fine level error |
|---:|---:|---:|
| 1.01 | 0.0098 | 0.0048 |
| 1.05 | 0.0098 | 0.0048 |
| 1.10 | 0.0100 | 0.0049 |
| 1.20 | 0.0101 | 0.0049 |
| 1.30 | 0.0103 | 0.0050 |
| 1.50 | 0.0106 | 0.0051 |
| 2.00 | 0.0113 | 0.0053 |
| 5.00 | 0.0139 | 0.0066 |

## C  RENDERING AND MESH EXTRACTION

To render the zero level set $f^{-1}(0)$ of a SDF $f$, two common strategies can be used: **sphere tracing** (ST) Hart et al. (1989), which directly traces rays through the SDF field, and **marching cubes** Lorensen & Cline (1987), which extracts an explicit mesh followed by standard rasterization. When the SDF is represented using M-plicits, both rendering approaches become more efficient.

We first introduce a **multiscale sphere tracing** scheme. Given a view ray $\gamma(t) = p_0 + tv$, with origin at point $p_0$, unit direction $v$, and intersecting the zero level set $f^{-1}(0)$, the standard sphere tracing (ST) approximates the first intersection point by iterating

$$p_{i+1} = p_i + v f(p_i)$$

along $\gamma$. However, querying a high-capacity neural SDF at each step can be computationally expensive. To reduce this cost, we leverage the multiscale SDF hierarchy $\{f_i\}$, using coarser networks to guide the early steps of the tracing process. Thanks to the nesting condition introduced in Eq. 3 of the paper, coarse levels can safely be used to trace offset surfaces before switching to finer levels closer to the surface. The ray initially traces the offset surface $f_1^{-1}(\delta_1)$ using $f_1$, proceeds to $f_2^{-1}(\delta_2)$ with $f_2$, and finally goes to the target surface $f_3^{-1}(0)$ using $f_3$. Each coarser level performs offset tracing via

$$p_{i+1} = p_i + v \left( f_j(p_i) - \delta_j \right),$$

which ensures convergence toward the true surface with minimal reliance on high-cost evaluations. Figure 11 illustrates this procedure, focusing on how the ray reaches $S_3$ by tracing within the neighborhood $\{|f_2| < \delta_2\}$. For neural SDF inference, we use the GEMM algorithm (Dongarra et al., 1990).

Importantly, if $\gamma \cap S_3 \neq \emptyset$, the multiscale ST approximates the first intersection point between $\gamma$ and $S_3$. This is guaranteed by the nesting condition, which implies that if $\gamma \cap S_3 \neq \emptyset$, then necessarily $\gamma \cap f_2^{-1}(\delta_2) \neq \emptyset$, and thus also $\gamma \cap f_1^{-1}(\delta_1) \neq \emptyset$. The values $\delta_i$ play a critical role in this process, as setting them appropriately helps avoid failures, as illustrated in Fig. 4 of the paper. Equation 5 (in the main paper) provides a principled definition of $\delta_i$, linking them to network training.

Finally, M-plicits also accelerates mesh extraction via **marching cubes**. We propose an adaptive grid inference strategy: we first evaluate the coarse SDF $f_1$ to cull grid vertices, and only evaluate finer SDFs for vertices inside the $\delta_1$-neighborhood. This yields efficient and focused SDF sampling, as depicted in Fig. 5 of the paper. Fig. 19 shows reconstructions using this approach.

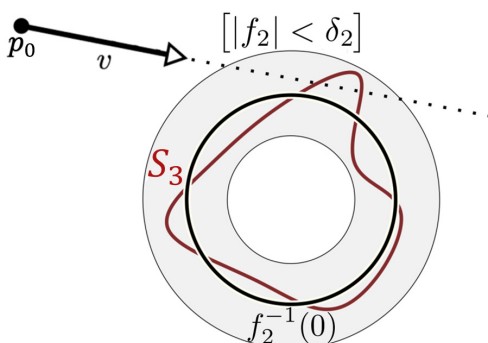
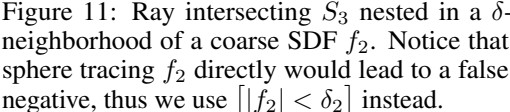
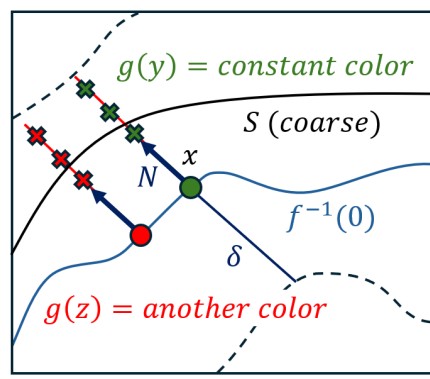

Figure 11: Ray intersecting $S_3$ nested in a $\delta$-neighborhood of a coarse SDF $f_2$. Notice that sphere tracing $f_2$ directly would lead to a false negative, thus we use $\left[|f_2| < \delta_2\right]$ instead.

Figure 12: Volumetric texture mapping. The texture $g$ should be constant along the normals $N$ near the coarse surface $S$ (red/green). Having such volumetric representation in the $\delta$-neighborhood ensures that $g$ can be assigned to any point in the coarse surface $S$.

## D NORMAL AND TEXTURE MAPPING

Let $S$ be a surface nested within a $\delta$-neighborhood of the zero level set of a neural SDF $f$, i.e., $S \subset \left[|f| \leq \delta\right]$. Assume $f$ is a finer neural SDF. Then, the *neural normal mapping* assigns to each point $p \in S$ the attribute

$$g(p) := \nabla f(p).$$

This corresponds to restricting $\nabla f$ to $S$, effectively transferring the normal of $f^{-1}(0)$ along the shortest path connecting it to $p$. Since $f$ is a signed distance function, the gradient $\nabla f$ remains constant along such paths.

We consider two cases. First, let $S$ be a triangle mesh. We use neural normal mapping to transfer detailed normals from the level sets of $f$ onto $S$. This approach is analogous to classical normal mapping, which typically relies on UV parameterizations. However, since our method is volumetric, such parameterizations are not required (see Fig. 7 in the paper, middle).

In the second case, let $S$ be the zero level set of a coarser neural SDF. Here, neural normal mapping allows us to bypass additional sphere tracing iterations (see Fig. 7 in the paper, left). In this case, surface extraction via marching cubes is not necessary.

Similarly, we define a neural network $g : \mathbb{R}^3 \to \mathcal{C}$ to encode a *texture* over the $\delta$-neighborhood of $f$, where the codomain $\mathcal{C}$ is the RGB color space. The attribute mapping associated with the triple $\{S, f, g\}$ is referred to as *neural texture mapping*.

To train the parameters $\phi$ of $g$, we use the following loss functional:

$$\mathcal{T}(\phi) = \int_{f^{-1}(0)} (g - \mathcal{q})^2 \, dx + \int_{\left[|f| \leq \delta\right]} \langle \nabla g, \nabla f \rangle^2 \, dx,$$

where the first term encourages $g$ to match the *ground-truth* texture $\mathcal{q}$, and the second term enforces consistency of $g$ along the gradient paths of $f$, regularizing the network within the $\delta$-neighborhood. Figure 12 illustrates the texture mapping scheme.

## E GEMM-BASED ANALYTICAL NORMAL CALCULATION FOR MLPS

We propose a GEMM-based analytical computation of normals, which are continuous and do not need auto-differentiation, resulting in smooth normals. To compute them, we recall that a MLP with $n - 1$ hidden layers has the following form $f(x) = W_n \circ h_{n-1} \circ \cdots \circ h_0(x) + b_n$, where $h_i(x_i) = \varphi(W_i x_i + b_i)$ is the i-layer. The *activation* $\varphi$ is applied on each coordinate of the linear map $W_i : \mathbb{R}^{N_i} \to \mathbb{R}^{N_{i+1}}$ translated by $b_i \in \mathbb{R}^{N_{i+1}}$. The gradient of $f$ is given using the *chain rule*:

$$\nabla f(x) = W_n \cdot \mathbf{J}h_{n-1}(x_{n-1}) \cdot \cdots \cdot \mathbf{J}h_0(x), \quad \text{with} \quad \mathbf{J}h_i(x_i) = W_i \odot \varphi'\left[a_i | \cdots | a_i\right] \qquad (6)$$

$\mathbf{J}$ is the *Jacobian*, $x_i := h_{i-1} \circ \cdots \circ h_0(x)$, $\odot$ is the *Hadamard* product, and $a_i = W_i(x_i) + b_i$. Eq. 6 is used in (Gropp et al., 2020; Novello et al., 2022) to compute the level set normals analytically.

We now use Eq. 6 to derive a GEMM-based algorithm for computing the normals ($\nabla f$) in real-time. The gradient $\nabla f$ is given by a sequence of matrix multiplications which is not appropriate for a GEMM setting because $\mathbf{J}h_0(x) \in \mathbb{R}^{3 \times N_1}$. The GEMM algorithm organizes the input points into a matrix, where its lines correspond to the points and its columns organize them and enable parallelism. We can solve this problem using three GEMMs, one for each normal coordinate. Therefore, each GEMM starts with a column of $\mathbf{J}h_0(x)$, eliminating one of the dimensions. The resulting multiplications can be asynchronous since they are completely independent.

The $j$-coord of $\nabla f$ is given by $G_n = W_n \cdot G_{n-1}$, where $G_{n-1}$ is given by iterating $G_i = \mathbf{J}h_i(x_i) \cdot G_{i-1}$, with the initial condition $G_0 = W_0[j] \odot \varphi'(a_0)$. The vector $W_0[j]$ denotes the $j$-column of $W_0$. We use a kernel and a GEMM to compute $G_0$ and $G_n$. For $G_i$ with $0 < i < n$, observe that

$$G_i = (W_i \odot \varphi' [a_i | \cdots | a_i]) \cdot G_{i-1} = (W_i \cdot G_{i-1}) \odot \varphi'(a_i).$$

The first equality comes from Eq. 6 and the second from a commutative property of the Hadamard product. The second expression needs fewer computations and is solved using a GEMM followed by a kernel.

Algorithm 1 presents the gradient computation for a batch of points. The input is a matrix $P \in \mathbb{R}^{3 \times k}$ with columns storing the $k$ points generated by the GEMM version of the sphere tracing algorithm. The output is a matrix $\nabla f_\theta(P) \in \mathbb{R}^{3 \times k}$, where its $j$-column is the gradient of $f_\theta$ evaluated at $P[j]$. Lines $2 - 5$ are responsible for computing $G_0$, Lines $6 - 11$ compute $G_{n-1}$, and Line 13 provides the result gradient $G_n$. Table 10 shows a comparison between this algorithm and automatic differentiation using PyTorch.

Table 10: Runtime comparison, in seconds, between Pytorch autograd and our algorithm to calculate the normals. Ours performs $2 \times$ faster.

| Model | Autograd | Ours | Resolution |
|---|---|---|---|
| Armadillo 256x2 | 0.007 | **0.003** | 512x512 |
| Armadillo 256x2 | 0.024 | **0.010** | 1024x1024 |
| Armadillo 256x3 | 0.010 | **0.005** | 512x512 |
| Armadillo 256x3 | 0.025 | **0.012** | 1024x1024 |
| Buddha 256x2 | 0.008 | **0.005** | 512x512 |
| Buddha 256x2 | 0.021 | **0.014** | 1024x1024 |
| Buddha 256x3 | 0.011 | **0.005** | 512x512 |
| Buddha 256x3 | 0.024 | **0.012** | 1024x1024 |
| Lucy 256x2 | 0.007 | **0.004** | 512x512 |
| Lucy 256x2 | 0.021 | **0.012** | 1024x1024 |
| Lucy 256x3 | 0.011 | **0.007** | 512x512 |
| Lucy 256x3 | 0.025 | **0.015** | 1024x1024 |

**ALGORITHM 1:** Normal computation

**Input:** neural SDF $f_\theta$, positions $P$
**Output:** Gradients $\nabla f_\theta(P)$
1 **for** $j = 0$ *to* 2 *(async)* **do**
2     using a GEMM: // Input Layer
3        $A_0 = W_0 \cdot P + b_0$
4     using a kernel:
5        $G_0 = W_0[j] \odot \varphi'(A_0)$; $P_0 = \varphi(A_0)$
    // Hidden layers
6     **for** *layer* $i = 1$ *to* $n - 1$ **do**
7        using GEMMs:
8           $A_i = W_i \cdot P_{i-1} + b_i$;
          $G_i = W_i \cdot G_{i-1}$
9        using a kernel:
10           $G_i = G_i \odot \varphi'(A_i)$; $P_i = \varphi(A_i)$
11     **end**
12     using a GEMM: // Output layer
13        $G_n = W_n \cdot G_{n-1}$
14 **end**

# F    SURFACE RECONSTRUCTION FROM POSED IMAGES

We demonstrate that M-plicits can be seamlessly integrated into image-based reconstruction pipelines, such as NeuS Wang et al. (2021). To this end, we replace the standard neural SDF used in NeuS with our multiscale SDF architecture, composed of two neural networks: a coarse-level network and a higher-resolution refinement network. We compare our modified pipeline against the baseline NeuS to evaluate the impact of our approach both quantitatively and qualitatively. Specifically, we implement the coarse network with 4 hidden layers of 128 neurons each, and the fine-level network with 5 hidden layers of 256 neurons. Despite having 37% fewer parameters than the original NeuS architecture, our multiscale approach achieves improved performance. Quantitatively, under the default volume rendering configuration, our method yields an average PSNR improvement of 3.74% across models from the DTU dataset Jensen et al. (2014). Table 11 summarizes the PSNR comparisons between our approach and the baseline NeuS.

Table 11: PSNR comparison between the baseline NeuS and our multiscale method on selected scans from the DTU dataset Jensen et al. (2014). Our method consistently improves reconstruction quality while using fewer network parameters. All values are reported in dB.

|      | Scan 24 | Scan 37 | Scan 40 | Scan 55 | Scan 63 | Avg. |
|------|---------|---------|---------|---------|---------|------|
| NeuS | 28.20   | 27.10   | 28.13   | 28.80   | 32.05   | 28.86 |
| Ours | 31.50   | 26.50   | 27.78   | 29.01   | 34.89   | 29.94 |

Figure 13 presents a comparison between NeuS and our multiscale variant on Scan 24 of the DTU dataset. The top row shows the reconstructed mesh geometry, including zoomed-in insets that highlight fine surface details. The bottom row displays renderings showing that ours provide reconstruction with sharp details. Our method yields improved geometric fidelity and cleaner surface reconstructions, especially in regions with architectural features.

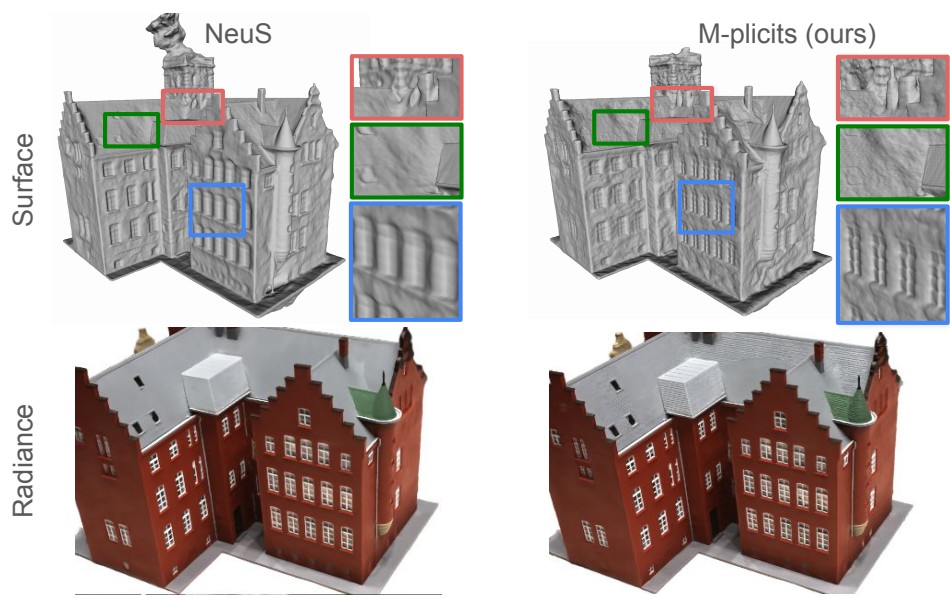

Figure 13: Comparison between baseline NeuS and our multiscale variant on Scan 24 of the DTU dataset. Top: extracted surface meshes with zoomed-in details. Bottom: rendered appearance. Our approach recovers finer geometric details, as evident in architectural structures and window boundaries.

## G  NEURAL IMPLICIT SURFACE EVOLUTION

Note that neural SDFs provide a smooth representation of a static scene. By adding an additional input coordinate, we can encode time into the representation. We leverage this approach to train dynamic evolutions of static neural SDFs, following the training schemes introduced in (Novello et al., 2023). Fig. 14 presents an example of interpolation between the Spot and Bob models using this method. Importantly, the implicit model handles topology changes, demonstrating that our representation can be integrated into differentiable pipelines. The visualization is in real-time (120 FPS) using an extension of our multiscale ST to dynamic SDFs.

### G.1  ADDITIONAL EXPERIMENTS

**Point cloud from images:**   Fig 15 shows our model trained with a point cloud reconstructed from an image. We use Depth Anything (Yang et al., 2024) to generate the depth of the pixels and use that depth to create the point cloud based on the view.

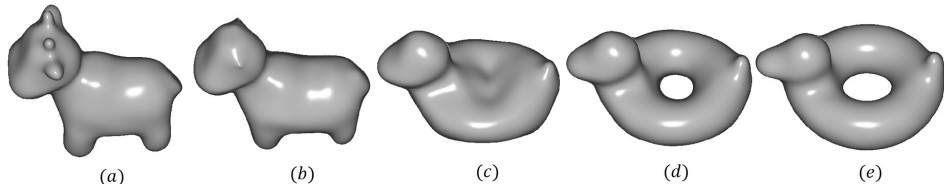

Figure 14: A dynamic multiscale SDF is trained using the pipeline from (Novello et al., 2023). Note the change in topology (c-d), which is challenging to handle using meshes. Also, octree/mesh-based approaches require generating a surface for each time, an overhead that our model avoids.

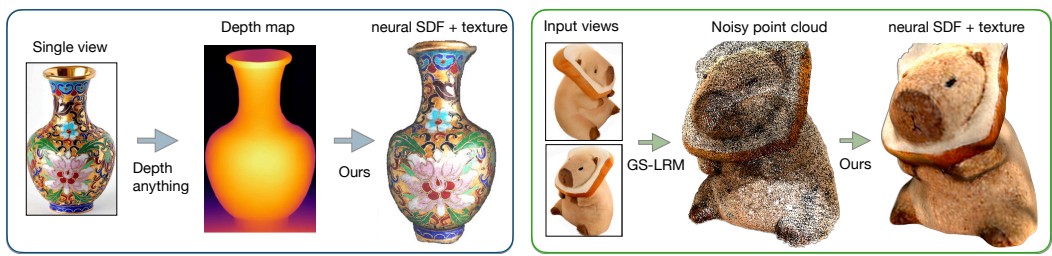

Figure 15: Training a textured SDF from images/noisy point cloud. On the left, our model (neural SDF + texture) is trained using the unprojection of a depth map, which is computed from a single view using Depth Anything. The resulting vase is rendered at 32.1 FPS. On the right, we show a reconstruction derived from a noisy point cloud, extracted from multiple views using GS-LRM (Zhang et al., 2024). By combining our method with this feed-forward 3D model (GS-LRM), we achieve fast reconstruction of the SDF with texture.

**Scene-scale point cloud.** We also evaluated our method on a scene-scale point cloud containing more than 10 million points. Figure 16 shows that M-plicits successfully reconstructs the entire scene across all scales (coarse, medium, and fine), whereas Instant-NGP fails in our tests using both 3 levels (coarse) and 16 levels (fine). The Chamfer distance further corroborates these observations: M-plicits achieves 2–3 orders of magnitude lower Chamfer distance compared to Instant-NGP.

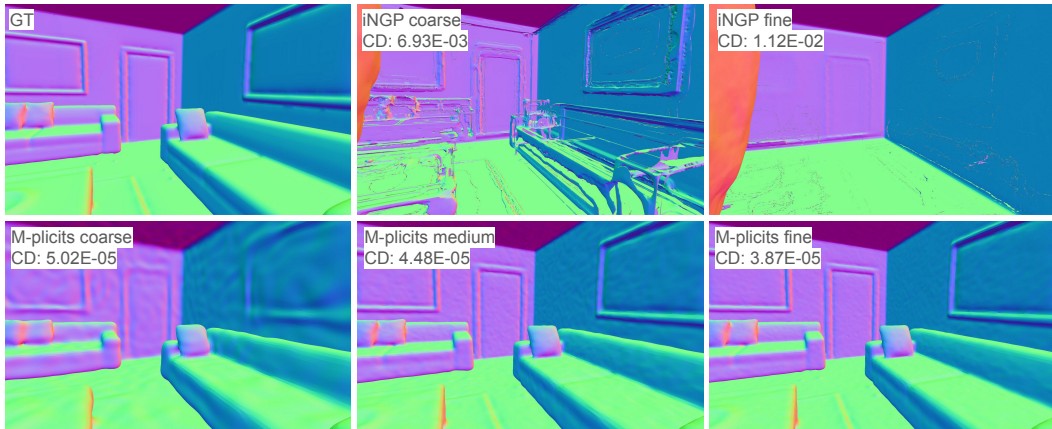

Figure 16: **Scene-scale reconstruction (10M+ points).** Comparison between Instant-NGP (coarse and fine) and M-plicits at three scales (coarse, medium, fine). Instant-NGP fails to reconstruct large regions of the geometry and introduces strong artifacts, even when increasing the number of levels. In contrast, M-plicits yields accurate and stable reconstructions at all scales, closely matching the ground-truth surface, as reflected in the reported Chamfer distances.

| **Coarse** | **Normal mapping** | **Multiscale ST** | **Baseline** |

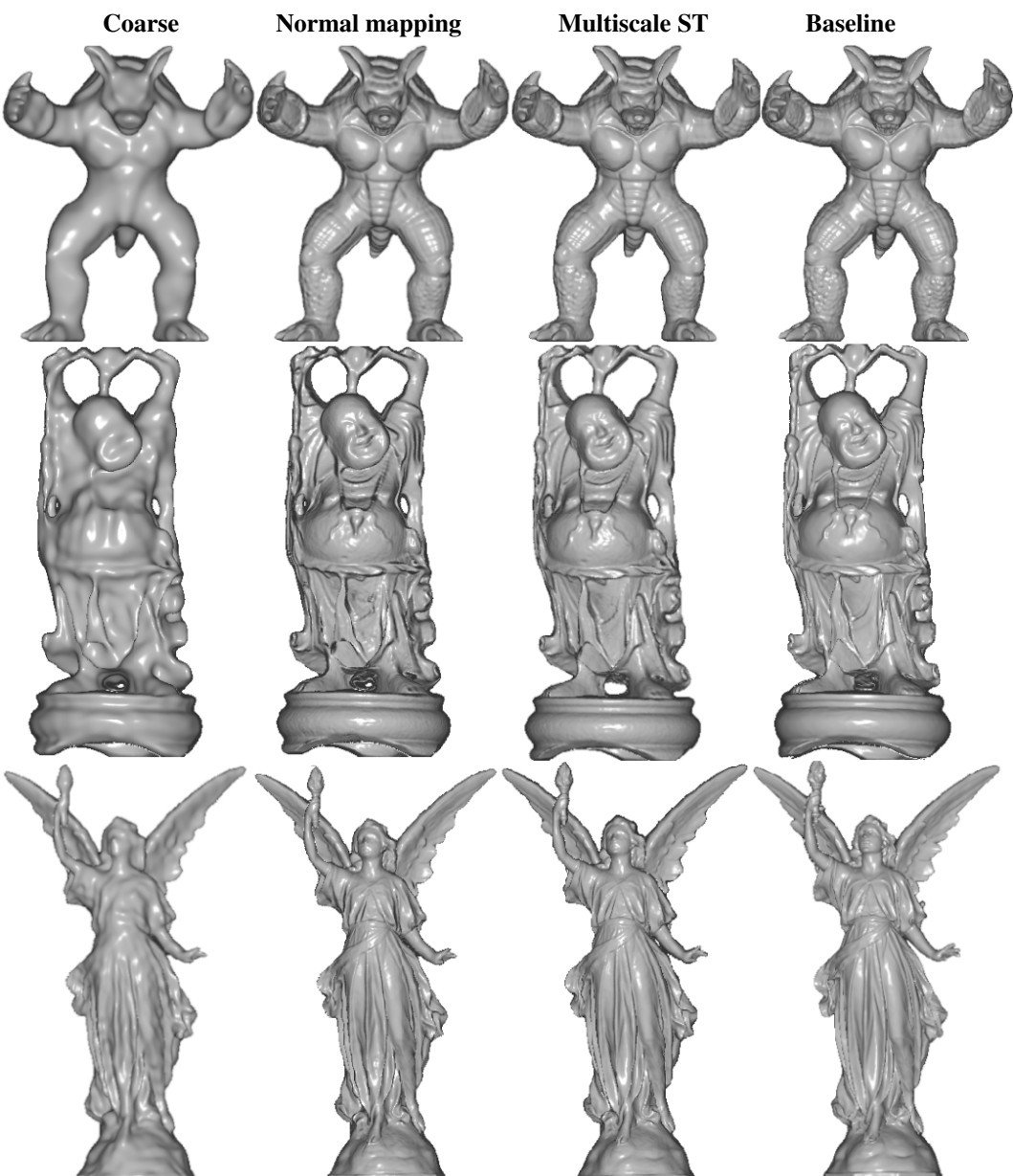

Figure 17: Comparison between our method and the SIREN baseline. The columns represent different configurations. From left to right: $(64, 2)$, $(64, 2) \triangleright (256, 2)$, and the baseline $(256, 4)$. The second column uses neural normal mapping and the third uses multiscale sphere tracing. Notice that fidelity is improved in the second column and the third column refines the results.

**Broader perceptual evaluation:** Fig. 17 shows a broader perceptual evaluation of the multiscale sphere tracing and the neural normal mapping using several models. Fig. 18 also shows the images we use to calculate the MSE to compare the neural texture mapping with the rendering baseline.

**Accelerated Marching Cubes qualitative evaluation:** Fig. 19 shows high-fidelity reconstructions computed using our acceleration for the marching cubes algorithm.

**Baseline** **Ours**

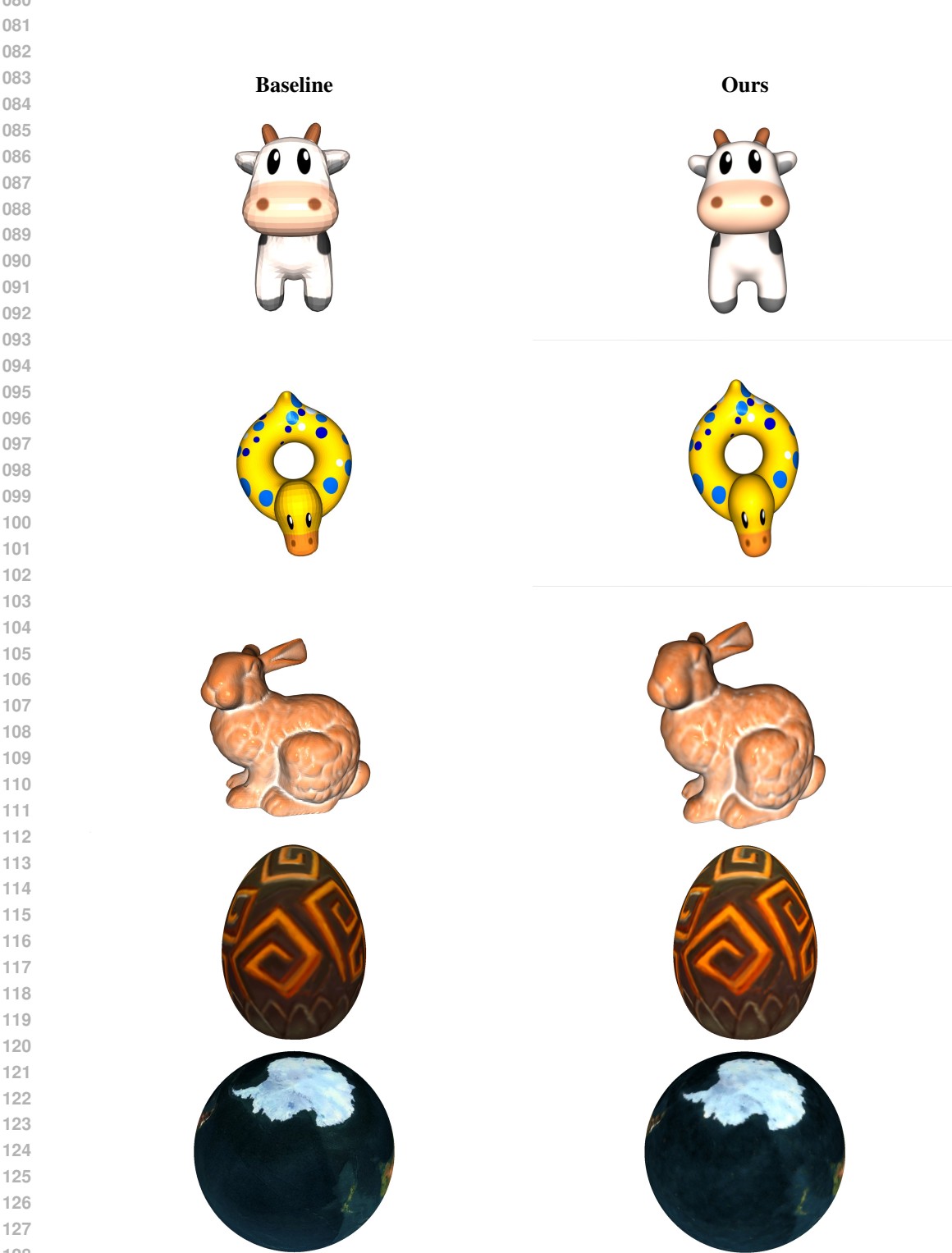

Figure 18: Images we use to calculate the MSE between the ground-truth textured meshes and our approach.

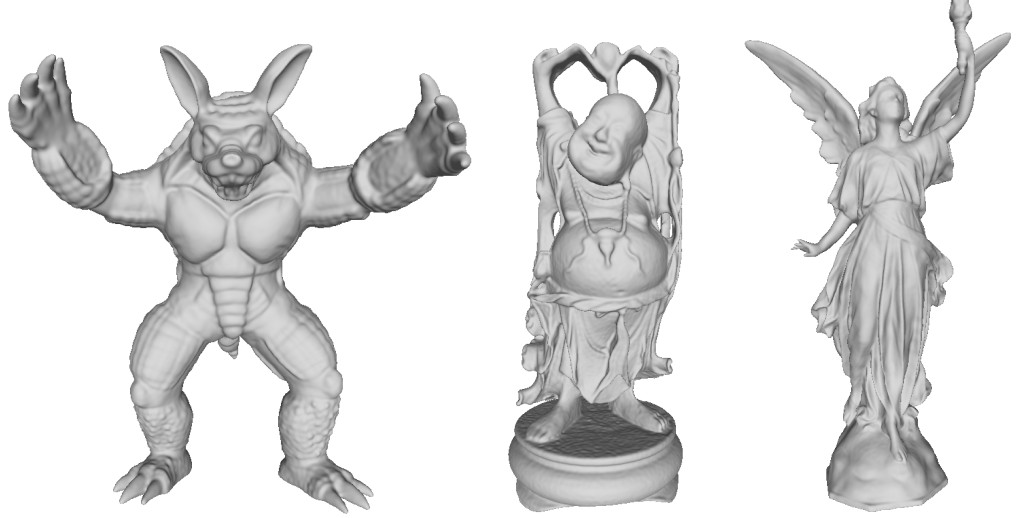

Figure 19: From left to right: Marching cubes reconstruction of Armadillo, Buddha and Lucy using our proposed grid culling method.

