# OpenReview forum: "Efficient Implicit Neural Surfaces via Multiscale Residuals and Nested Training"
_ICLR.cc/2026/Conference — Submitted to ICLR 2026_

### Official Review · Reviewer_o828 · 2025-10-28

**Soundness:** 3
**Presentation:** 3
**Contribution:** 3
**Rating:** 6
**Confidence:** 3

**Summary:**

This paper addresses a major problem with implicit neural surfaces (like SIREN): they look great but are extremely slow to render or mesh. This slowness comes from using a single, giant neural network for the entire shape.

The authors propose M-plicits, a new method that splits the network into multiple scales. A very small, fast "base" network learns the coarse, overall shape. Then, other "residual" networks are added on top to handle the fine-grained details.

The key idea is in the training: these detail networks are only trained in a very narrow band right around the object's surface, not everywhere. This concentrates the network's power exactly where it's needed.

As a result, the method is much faster. When rendering, any query far from the object only needs to run through the tiny base network, which is extremely cheap. The paper shows this approach achieves real-time rendering (beating NGLOD) and dramatically faster meshing (beating IDF) while keeping the model size incredibly small.

**Strengths:**

1. Solves an important problem: The paper makes pure implicit models, which are usually very slow, practical and fast. This is a really useful contribution.

2. Clever training method: The core idea of "nested training" (only training details near the surface) is simple but works extremely well. The ablation study shows this is a key reason for the high quality and it's not just a small trick.

3. Extremely fast and efficient: This is the best part of the paper. The results are impressive: It's incredibly fast at making meshes, blowing other methods (like IDF) out of the water (e.g., 1 second vs. 67 seconds). It can render high-quality shapes in real-time (70 FPS), which is even faster than the low-quality version of NGLOD. The models are tiny (very small file size), which is great.

5. Good experiments: The authors did a good job with ablation studies to prove their different ideas work, like the custom normal calculation.

**Weaknesses:**

1. Robustness of the Neighborhood $\delta$: The neighborhood size $\delta$ is a critical hyperparameter. While the paper proposes an adaptive strategy (Eq. 5) and a brief ablation (Table 8), its robustness is not exhaustively discussed. The implementation detail of "normalizing all point clouds to the unit sphere" may be hiding sensitivity to scale. How does the adaptive strategy for $\delta$ perform on point clouds of arbitrary or varying scales (e.g., in meters vs. centimeters) without this normalization step?

2. Missing Ablation on Number of Levels: The paper's core is a "multiscale" design, yet it primarily shows 2 and 3-level results. A dedicated ablation study on the number of levels (k) is missing. A systematic study is needed to show the relationship between k, the final reconstruction quality (CD), parameter scaling, and inference speed. This is crucial for understanding the true trade-off curve and demonstrating the scalability of the proposed multiscale architecture.

3. Generalization to Scene-Scale Point Clouds: I noted the strong results on DTU scenes in Appendix F. However, this experiment integrates M-plicits into NeuS, which relies on multi-view images and volume rendering . This is a fundamentally different task and input modality from the main paper's focus on surface reconstruction from oriented point clouds (Sec 3, Stanford dataset). Does this imply that the core NNT method does not scale directly to large-scale, sparse, or noisy scene point clouds (e.g., from LiDAR scans) and must instead rely on image-based volume rendering to handle scene-scale tasks?

**Questions:**

See Weaknesses section.

---

> ### Author Response · Authors · 2025-11-27
>
> We thank the reviewer for the encouraging comments regarding our clever training strategy, efficiency, the problem importance and ablation experiments. We will address your concerns below.
>
> > Robustness of the Neighborhood $\delta$: The neighborhood size $\delta$ is a critical hyperparameter. While the paper proposes an adaptive strategy (Eq. 5) and a brief ablation (Table 8), its robustness is not exhaustively discussed. The implementation detail of "normalizing all point clouds to the unit sphere" may be hiding sensitivity to scale. How does the adaptive strategy for  perform on point clouds of arbitrary or varying scales (e.g., in meters vs. centimeters) without this normalization step?
>
> Equation 5 depends on three quantities:  (1) the previously trained SDF level,  (2) the parameter $ε$, and  (3) the input samples $x_j$ (which lie on or near the target zero level set).
>
>
> Table 9 provides an ablation on the ε parameter, showing that the method is not overly sensitive to this choice.
>
>
> Beyond this ablation, we include additional experiments to test the robustness of these parameters under challenging conditions—specifically, noisy point clouds (Table 4 and Figure 10) and a scene-scale point cloud with more than 10M points (Figure 16). Please see the subsequent answers for details.
>
>
> > Missing Ablation on Number of Levels: The paper's core is a "multiscale" design, yet it primarily shows 2 and 3-level results. A dedicated ablation study on the number of levels (k) is missing. A systematic study is needed to show the relationship between k, the final reconstruction quality (CD), parameter scaling, and inference speed. This is crucial for understanding the true trade-off curve and demonstrating the scalability of the proposed multiscale architecture.
>
>
> We performed an additional ablation test on the number of levels, increasing the number of levels to 6. We present the Chamfer distance of each level in the following table. We used the Buddha of the Stanford dataset and a subset of the Thing10k. In all cases of this experiment, the Chamfer distance monotonically decreases up to level 3 (equivalent to our proposed fine level). Those values corroborate our choice of using 3 levels in our representation. It is important to note that our approach does not claim to be a level-of-detail method, but a representation with fixed different scales that enable fast and accurate distance and attribute (normals and texture color) queries.
>
>
> | Model | Lvl 1 CD | Lvl 2 CD | Lvl 3 CD | Lvl 4 CD | Lvl 5 CD | Lvl 6 CD |
> |---------|----------:|----------:|----------:|----------:|----------:|----------:|
> | Buddha | 2.90E-03 | 2.29E-03 | **2.27E-03** | 2.29E-03 | 3.32E-03 | 2.17E-02 |
> | 44234 | 1.41E-03 | 1.08E-03 | **1.06E-03** | 4.65E-02 | 6.06E-02 | 6.18E-02 |
> | 64764 | 2.13E-02 | 2.10E-02 | **2.08E-02** | 4.88E-02 | 3.77E-02 | 3.53E-02 |
> | 68381 | 8.77E-03 | 8.58E-03 | **8.46E-03** | 3.17E-02 | 3.48E-02 | 3.58E-02 |
> | 72870 | 2.83E-03 | 2.06E-03 | **1.85E-03** | 3.25E-02 | 4.61E-02 | 5.18E-02 |
> | 73075 | 1.43E-03 | 1.22E-03 | **1.07E-03** | 1.81E-02 | 6.93E-02 | 9.59E-02 |
> | 77245 | 3.62E-02 | 3.57E-02 | **3.54E-02** | 7.83E-02 | 5.47E-02 | 5.13E-02 |
> | 354371 | 3.25E-03 | 2.80E-03 | **2.79E-03** | 3.98E-03 | 3.68E-02 | 2.94E-02 |
>
>
> > Generalization to Scene-Scale Point Clouds: I noted the strong results on DTU scenes in Appendix F. However, this experiment integrates M-plicits into NeuS, which relies on multi-view images and volume rendering . This is a fundamentally different task and input modality from the main paper's focus on surface reconstruction from oriented point clouds (Sec 3, Stanford dataset). Does this imply that the core NNT method does not scale directly to large-scale, sparse, or noisy scene point clouds (e.g., from LiDAR scans) and must instead rely on image-based volume rendering to handle scene-scale tasks?
>
> We performed additional tests on a **scene-scale point cloud** dataset with more than 10.000.000 points. Figure 16 in the revised paper shows that our approach is able to correctly reconstruct the scene in all proposed scales (coarse, medium, and fine) while Instant-NGP fails in our tests with 3 levels (coarse) and 16 levels (fine). We also present metrics (Chamfer, Hausdorff, and IoU) that corroborate those results in the following table. We achieve 2 or 3 orders of magnitude better Chamfer distance and one order of magnitude better IoU.
>
> | Method | Chamfer (⬇) | Hausdorff (⬇) | IoU (⬆) |
> |-------------------|-------------:|---------------:|-----------:|
> | Ours (coarse) | 5.02E-05 | **1.11E-02** | 2.46E-01 |
> | Ours (medium) | 4.48E-05 | 1.14E-02 | 2.69E-01 |
> | Ours (fine) | **3.87E-05** | **1.11E-02** | **3.83E-01** |
> | iNGP (3 levels) | 6.93E-03 | 5.98E-02 | 3.05E-02 |
> | iNGP (16 levels) | 1.12E-02 | 7.57E-02 | 2.44E-02 |

---

> ### Author Response · Authors · 2025-11-27
>
> We also evaluated the robustness of our approach to noise. In that experiment, all vertices are randomly perturbed in the direction of the normal by at most 1.0% of the model bounding box.  Figure 10 in the revised paper shows the results for the Lucy model. The coarse level of M-plicits acts as a low-pass filter, removing most of the high-frequency noise and providing a clean geometric prior that benefits the subsequent residual levels. In contrast, Instant-NGP (coarse/fine) fails to denoise the input and produces highly irregular surfaces. We also present metrics (Chamfer, Hausdorff, and IoU) for 3 models. The quantitative and qualitative data shows that our approach is very robust to noise.
>
> | Model | Chamfer (⬇) | Hausdorff (⬇) | IoU (⬆) |
> |-------------------------------|-------------:|---------------:|-----------:|
> | Armadillo (ours coarse) | 1.83E-03 | 1.18E-02 | 5.68E-02 |
> | Armadillo (ours medium) | **5.86E-05** | **7.18E-04** | **4.89E-01** |
> | Armadillo (ours fine) | 2.37E-03 | 1.21E-02 | 5.09E-02 |
> | Armadillo (iNGP coarse) | 1.75E-02 | 1.05E-01 | 3.54E-02 |
> | Armadillo (iNGP fine) | 1.93E-02 | 9.49E-02 | 3.34E-02 |
> | Asian Dragon (ours coarse) | 3.49E-04 | 1.55E-02 | 2.97E-01 |
> | Asian Dragon (ours medium) | 2.02E-04 | 4.10E-03 | 3.70E-01 |
> | Asian Dragon (ours fine) | **1.69E-04** | **2.76E-03** | **4.00E-01** |
> | Asian Dragon (iNGP coarse) | 2.35E-02 | 2.21E-01 | 4.70E-03 |
> | Asian Dragon (iNGP fine) | 1.62E-02 | 1.52E-01 | 6.68E-03 |
> | Lucy (ours coarse) | 3.33E-04 | 2.21E-02 | 3.02E-01 |
> | Lucy (ours medium) | 4.39E-05 | **8.31E-04** | 5.18E-01 |
> | Lucy (ours fine) | **3.45E-05** | 8.54E-04 | **5.98E-01** |
> | Lucy (iNGP coarse) | 2.36E-02 | 1.36E-01 | 0.00E+00 |
> | Lucy (iNGP fine) | 1.12E-02 | 7.57E-02 | 1.00E-03 |

---

### Official Review · Reviewer_wPDQ · 2025-10-29

**Soundness:** 3
**Presentation:** 4
**Contribution:** 2
**Rating:** 6
**Confidence:** 3

**Summary:**

This paper introduces M-plicits, which models the surface SDF using a base network and three smaller MLPs (coarse, medium, and fine) to capture surface details at different scales. The final SDF value is obtained by summing the predictions of all networks, with each finer MLP providing residual corrections to the base SDF, leading to progressively refined outputs.
The key contribution of these multiple MLPs lies in concentrating the modeling accuracy near the surface, enabling efficient high-resolution mesh extraction and real-time rendering. However, the residual refinement, though aimed at capturing high-frequency signals (geometric details), provides limited improvement in surface reconstruction accuracy.

**Strengths:**

1. The paper proposes multiple MLPs to compute residual SDF values of points near the surface at different scales (base, coarse, medium, and fine). This hierarchical design progressively narrows the query region from distant to near-surface points, reducing unnecessary evaluations far from the surface.
2. Efficient sampling strategies are introduced to ensure dense coverage of regions close to the surface, resulting in more effective training of the residual networks and faster inference for mesh extraction and sphere tracing.
3. An additional neural network is introduced alongside the multiple MLPs for SDFs to represent texture attributes.

**Weaknesses:**

1. The contribution of the multiscale residual MLPs to surface reconstruction accuracy appears limited. For example, in Table 1, for the Thai Statue reconstruction, the results of “Ours (fine)” and “Ours (coarse)” show no significant improvement, and for Armadillo, “Ours (coarse)” even performs better than “Ours (fine)”.
2. Results on only four shapes in Table 1 are not compelling to support the claim of high-quality reconstruction. It would strengthen the paper to (i) include additional evaluation metrics beyond Chamfer Distance (CD) and (ii) provide visual comparisons of reconstructed meshes across different methods.
3. How does a single base SDF network perform in terms of reconstruction accuracy if trained with a comparable number of parameters to the combined residual SDF networks? This comparison could help isolate the benefit contributed by the multiscale decomposition, which so far appears to mainly improve computational efficiency by reducing the query space near the surface.
4. It would be helpful to visualize the reconstructed surfaces of the proposed method at different residual levels to highlight potential improvements in surface detail.

**Questions:**

(1) How many points are sampled for the training of neural SDF at different scales?

(2) In the neural normal mapping (Lines 282–283), only the input derivatives of the finer-level neural SDF are used. What does the gradient of the combined SDF with respect to $x$ look like, i.e., $\nabla_x (f_1(x) + r_1(x) + \dots + r_i(x))$?

---

> ### Author Response · Authors · 2025-11-27
>
> We thank the reviewer for your detailed comments and questions. We address those below.
>
> > The contribution of the multiscale residual MLPs to surface reconstruction accuracy appears limited. For example, in Table 1, for the Thai Statue reconstruction, the results of “Ours (fine)” and “Ours (coarse)” show no significant improvement, and for Armadillo, “Ours (coarse)” even performs better than “Ours (fine)”.
>
> > Results on only four shapes in Table 1 are not compelling to support the claim of high-quality reconstruction. It would strengthen the paper to (i) include additional evaluation metrics beyond Chamfer Distance (CD) and (ii) provide visual comparisons of reconstructed meshes across different methods.
>
> We present additional experimental results, including the IoU of the Stanford dataset. Except for the Armadillo, all other models provide better metrics for the fine cases. It is also important to note that the other methods may also suffer from worse metrics for the fine cases. However, qualitative experiments of our approach (Figure 16 of the revised manuscript) show that the fine level of our approach (Multiscale ST column) is perceptually better than the coarse level. For example, Instant NGP presents similar values between the coarse and fine models for the Armadillo and worse case for the Lucy.
>
>
> | Model | IoU (⬆) |
> |--------------------------------|------------:|
> | Armadillo (ours coarse) | **2.60E-01** |
> | Armadillo (ours fine) | 1.91E-01 |
> | Armadillo (iNGP coarse) | 2.34E-02 |
> | Armadillo (iNGP fine) | 2.33E-02 |
> | Asian Dragon (ours coarse) | 7.14E-01 |
> | Asian Dragon (ours fine) | **8.81E-01** |
> | Asian Dragon (iNGP coarse) | 8.47E-03 |
> | Asian Dragon (iNGP fine) | 7.87E-03 |
> | Lucy (ours coarse) | 3.46E-01 |
> | Lucy (ours fine) | **6.63E-01** |
> | Lucy (iNGP coarse) | 1.96E-03 |
> | Lucy (iNGP fine) | 2.07E-03 |
> | Thai Statue (ours coarse) | 7.54E-02 |
> | Thai Statue (ours fine) | **9.20E-02** |
> | Thai Statue (iNGP coarse) | 1.17E-02 |
> | Thai Statue (iNGP fine) | 1.26E-02 |
>
>
> We performed an additional ablation test on the number of levels on other surfaces, increasing the number of levels to 6. We present the Chamfer distance of each level in the following table. We used the Buddha of the Stanford dataset and a subset of the Thing10k. In all cases of this experiment, the Chamfer distance monotonically decreases up to level 3 (equivalent to our proposed fine level). Those values corroborate our choice of using 3 levels in our representation. It is important to note that our approach does not claim to be a level-of-detail method, but a representation with fixed different scales that enable fast and accurate distance and attribute (normals and texture color) queries.
>
>
> | Model | Lvl 1 CD | Lvl 2 CD | Lvl 3 CD | Lvl 4 CD | Lvl 5 CD | Lvl 6 CD |
> |---------|----------:|----------:|----------:|----------:|----------:|----------:|
> | Buddha | 2.90E-03 | 2.29E-03 | **2.27E-03** | 2.29E-03 | 3.32E-03 | 2.17E-02 |
> | 44234 | 1.41E-03 | 1.08E-03 | **1.06E-03** | 4.65E-02 | 6.06E-02 | 6.18E-02 |
> | 64764 | 2.13E-02 | 2.10E-02 | **2.08E-02** | 4.88E-02 | 3.77E-02 | 3.53E-02 |
> | 68381 | 8.77E-03 | 8.58E-03 | **8.46E-03** | 3.17E-02 | 3.48E-02 | 3.58E-02 |
> | 72870 | 2.83E-03 | 2.06E-03 | **1.85E-03** | 3.25E-02 | 4.61E-02 | 5.18E-02 |
> | 73075 | 1.43E-03 | 1.22E-03 | **1.07E-03** | 1.81E-02 | 6.93E-02 | 9.59E-02 |
> | 77245 | 3.62E-02 | 3.57E-02 | **3.54E-02** | 7.83E-02 | 5.47E-02 | 5.13E-02 |
> | 354371 | 3.25E-03 | 2.80E-03 | **2.79E-03** | 3.98E-03 | 3.68E-02 | 2.94E-02 |
>
>
> We performed additional tests on a *scene-scale point cloud* dataset with more than 10.000.000 points. Figure 16 in the revised paper shows that our approach is able to correctly reconstruct the scene in all proposed scales (coarse, medium, and fine) while Instant-NGP fails in our tests with 3 levels (coarse) and 16 levels (fine). We also present metrics (Chamfer, Hausdorff, and IoU) that corroborate those results in the following table. We achieve 2 or 3 orders of magnitude better Chamfer distance and one order of magnitude better IoU.
>
>
> | Method | Chamfer (⬇) | Hausdorff (⬇) | IoU (⬆) |
> |-------------------|-------------:|---------------:|-----------:|
> | Ours (coarse) | 5.02E-05 | **1.11E-02** | 2.46E-01 |
> | Ours (medium) | 4.48E-05 | 1.14E-02 | 2.69E-01 |
> | Ours (fine) | **3.87E-05** | **1.11E-02** | **3.83E-01** |
> | iNGP (3 levels) | 6.93E-03 | 5.98E-02 | 3.05E-02 |
> | iNGP (16 levels) | 1.12E-02 | 7.57E-02 | 2.44E-02 |

---

> ### Author Response · Authors · 2025-11-27
>
> We also evaluated the robustness of our approach to noise. In that experiment, all vertices are randomly perturbed in the direction of the normal by at most 1.0% of the model bounding box.  Figure 10 in the revised paper shows the results for the Lucy model. The coarse level of M-plicits acts as a low-pass filter, removing most of the high-frequency noise and providing a clean geometric prior that benefits the subsequent residual levels. In contrast, Instant-NGP (coarse/fine) fails to denoise the input and produces highly irregular surfaces. We also present metrics (Chamfer, Hausdorff, and IoU) for 3 models. The quantitative and qualitative data shows that our approach is very robust to noise.
>
> | Model | Chamfer (⬇) | Hausdorff (⬇) | IoU (⬆) |
> |-------------------------------|-------------:|---------------:|-----------:|
> | Armadillo (ours coarse) | 1.83E-03 | 1.18E-02 | 5.68E-02 |
> | Armadillo (ours medium) | **5.86E-05** | **7.18E-04** | **4.89E-01** |
> | Armadillo (ours fine) | 2.37E-03 | 1.21E-02 | 5.09E-02 |
> | Armadillo (iNGP coarse) | 1.75E-02 | 1.05E-01 | 3.54E-02 |
> | Armadillo (iNGP fine) | 1.93E-02 | 9.49E-02 | 3.34E-02 |
> | Asian Dragon (ours coarse) | 3.49E-04 | 1.55E-02 | 2.97E-01 |
> | Asian Dragon (ours medium) | 2.02E-04 | 4.10E-03 | 3.70E-01 |
> | Asian Dragon (ours fine) | **1.69E-04** | **2.76E-03** | **4.00E-01** |
> | Asian Dragon (iNGP coarse) | 2.35E-02 | 2.21E-01 | 4.70E-03 |
> | Asian Dragon (iNGP fine) | 1.62E-02 | 1.52E-01 | 6.68E-03 |
> | Lucy (ours coarse) | 3.33E-04 | 2.21E-02 | 3.02E-01 |
> | Lucy (ours medium) | 4.39E-05 | **8.31E-04** | 5.18E-01 |
> | Lucy (ours fine) | **3.45E-05** | 8.54E-04 | **5.98E-01** |
> | Lucy (iNGP coarse) | 2.36E-02 | 1.36E-01 | 0.00E+00 |
> | Lucy (iNGP fine) | 1.12E-02 | 7.57E-02 | 1.00E-03 |
>
>
>
> > How does a single base SDF network perform in terms of reconstruction accuracy if trained with a comparable number of parameters to the combined residual SDF networks? This comparison could help isolate the benefit contributed by the multiscale decomposition, which so far appears to mainly improve computational efficiency by reducing the query space near the surface.
>
> > It would be helpful to visualize the reconstructed surfaces of the proposed method at different residual levels to highlight potential improvements in surface detail.
>
> We present a qualitative comparison between a SIREN baseline, which is equivalent to using a single base SDF in our approach with a comparable number of parameters, in Figure 16 of the revised manuscript. In that Figure, the Multiscale ST column presents our fine model and the Baseline column presents the SIREN single SDF. The Normal Mapping column represents the normals of the fine level mapped into the coarse level. Additionally, Table 2 in the revised manuscript shows that using our approach results in considerable rendering speedups, ranging from 2X to 8.5X.
>
> > How many points are sampled for the training of neural SDF at different scales?
>
> For the coarse level, we use the entire point cloud, for the medium we use 75k, and for all levels beyond, we use 60k points per batch, due to memory constraints
>
>
> > In the neural normal mapping (Lines 282–283), only the input derivatives of the finer-level neural SDF are used. What does the gradient of the combined SDF with respect to $x$ look like, i.e., $\nabla_x(f_1(x) + r_1(x) + \cdots + r_i(x))$?
>
> In our neural normal mapping, the goal is to transfer the higher-frequency normals from finer levels to the coarser ones. Recall that each refined SDF is defined as:
>
> $$f_2 = f_1 + r_1$$
>
> $$f_3 = f_2 + r_2 = f_1 + r_1 + r_2.$$
>
> Importantly, **the residuals $r_i$ are not SDFs**; only the *combined* functions $f_i + r_i$ are trained to satisfy the SDF property and therefore provide increasingly accurate approximations of the ground-truth normals.
>
> Regarding the gradient of the combined SDF, we simply apply linearity of differentiation:
>
> $$
> \nabla_x f_2(x)  = \nabla_x \left( f_1(x) + r_1(x) \right)
>                = \nabla_x f_1(x) + \nabla_x r_1(x),
> $$
>
> $$
> \nabla_x f_3(x) = \nabla_x \left( f_1(x) + r_1(x) + r_2(x) \right)
>                = \nabla_x f_1(x) + \nabla_x r_1(x) + \nabla_x r_2(x).
> $$
>
> Thus, the gradient of the combined SDF at level \(i\) is simply the **sum of the gradients of all previous components**.

---

### Official Review · Reviewer_PWMZ · 2025-10-31

**Soundness:** 3
**Presentation:** 2
**Contribution:** 2
**Rating:** 4
**Confidence:** 4

**Summary:**

The M-plicits framework introduces a novel architectural and training paradigm for INRs of surfaces, specifically designed to address a fundamental challenge in the field: the inherent tension between representational capacity and computational efficiency. Traditional approaches that employ a single, large MLP to model a surface as the zero-level set of a SDF often achieve high fidelity at the cost of prohibitive inference times. M-plicits re-architects the INR to overcome this limitation by concentrating its modeling power and computational resources specifically in the regions of high geometric complexity—near the surface itself.

**Strengths:**

The strength of M-plicits lies in its pragmatic and geometrically-grounded design. provides a system that is simple, robust, and effective for the task of reconstructing and rendering high-fidelity surfaces in real time. Its ability to simultaneously achieve top-tier reconstruction quality, the highest rendering framerates, and the most compact model sizes, as demonstrated in its empirical evaluation, marks a substantial step forward.

**Weaknesses:**

The paper's core weakness is its limited conceptual novelty. Its central ideas:the multiscale residual architecture, and the multi-resolution sphere tracing (cf. existing techniques) are not entirely novel.

**Questions:**

1) Please discuss your multi-scale sphere tracing algorithm in the context of existing coarse-to-fine SDF rendering techniques
2) Please clarify the fundamental conceptual difference between your "nested neighborhood training" and the "cascaded optimization" used in BANF.
3) In Figure8,why the middle level is 96 not 128

---

> ### Author Response · Authors · 2025-11-27
>
> We thank the reviewer for the feedback. We appreciate the recognition of the strengths of the paper, particularly the pragmatic design, strong empirical results, and real-time rendering with a compact INR. Below we address each concern in detail.
>
> > The paper's core weakness is its limited conceptual novelty. Its central ideas:the multiscale residual architecture, and the multi-resolution sphere tracing (cf. existing techniques) are not entirely novel.
>
> Our contribution is not simply the use of multiscale components, but the specific combination of a multiscale residual architecture with a nesting-neighborhood training scheme. This coupling is central to the method’s effectiveness. Beyond improving reconstruction quality, this design enables a set of capabilities (which are also INR contributions) that are not simultaneously achievable with previous methods:
>
> * Sphere tracing for real-time rendering (Table 2).
> * Fast analytical normals computed without backpropagation or autograd—requiring only a forward pass through the network.
> * Accelerated marching cubes for surface extraction (Table 3), achieving up to a 5× speedup over a baseline SIREN.
> * Texture support, allowing high-quality textured surfaces without degrading geometry (Figure 9).
>
> Existing multiresolution INRs, such as BANF, NGLOD, and Instant-NGP, are grid-dependent multi-resolution INRs. These approaches typically:
>
> 1) suffer from discontinuous normals at voxel boundaries,
>
> 2) require substantial memory, and
>
> 3) do not support real-time rendering (except Instant-NGP). Even Instant-NGP exhibits important limitations in SDF reconstruction, struggling with noisy point clouds (Figure 10) and failing on large-scale scenes (Figure 16), whereas M-plicits remain stable and accurate in both settings.
>
> > Please discuss your multi-scale sphere tracing algorithm in the context of existing coarse-to-fine SDF rendering techniques
>
> To the best of our knowledge, NGLOD is the only prior work that implements an adaptive coarse-to-fine sphere-tracing strategy for neural SDFs. Their acceleration relies on a sparse octree hierarchy with trilinear interpolation inside voxels and ray–AABB traversal between them. However, NGLOD’s early-level representations have very low geometric fidelity, and the first level is the only one that can be rendered in real-time in their implementation (see Figure 6 in the paper).
>
> Moreover, NGLOD’s coarse SDF may not contain the fine surface in its neighborhood, leading to missed intersections (“false negatives”). A key component of M-plicits is the nested SDF training scheme, which explicitly prevents this failure mode by ensuring that every level provides a valid coarse approximation of the final SDF.
>
> Importantly, we introduce a multiscale sphere-tracing algorithm that, when combined with our nesting condition, enables both real-time rendering and high-quality surface reconstruction.
>
> > Please clarify the fundamental conceptual difference between your "nested neighborhood training" and the "cascaded optimization" used in BANF.
>
> The cascaded optimization in BANF supervises every level directly using the ground truth, progressing from coarse to fine. Each level is trained sequentially, and every scale receives full-domain supervision during its stage.
>
> In contrast, M-plicits also trains the base SDF using standard neural-SDF supervision, but the key difference lies in our nested neighborhood training. New residual levels are not trained over the entire domain; instead, each level is optimized only within a small neighborhood around the zero-level set of the previous level (with radius defined in Eq. 5). This localized supervision ensures that each level becomes a valid coarse approximation of the next, concentrates modeling capacity where it matters most—near the surface—and prevents spurious artifacts in unsupervised regions (see the second row of Fig. 8).
>
> Additionally, because this neighborhood is sampled using our new surface-aware sampling strategy (Sec. 3.2), the nested scheme further improves reconstruction accuracy and stability.
>
> > In Figure8, why the middle level is 96 not 128
>
> That was a typo indeed. We use 128 neurons for the medium level. We fixed that in the revised paper.

---

### Official Review · Reviewer_ctkd · 2025-10-31

**Soundness:** 3
**Presentation:** 3
**Contribution:** 2
**Rating:** 4
**Confidence:** 4

**Summary:**

This paper proposes to use multi-Scale residuals for efficient learning Implicit Neural Representations (INRs) of 3D surfaces. The authors propose to replace the conventional single large MLP with a cascaded architecture employing Multi-Scale Residuals, where each subsequent, typically smaller MLP learns a residual correction focusing on progressively higher-frequency geometric detail. To this end, they introduce Nested Training, a sequential optimization scheme where each residual network scale is trained by supervising only near the coarser converged zero-level set, which significantly accelerates the overall training process. Furthermore, the authors further introduce a GEMM-accelerated normal computation method and an efficient mesh extraction pipeline that  fully supports normal and texture mapping by utilizing the multi-scale features. Experiments demonstrate that the method achieves superior or comparable geometric accuracy to state-of-the-art baselines while offering significant improvements in both training and inference efficiency.

**Strengths:**

The submission is clearly written and well-organized, presenting a technically sound pipeline. The step-by-step introduction of the
cascaded architecture and training scheme is easy to follow. The multi-scale residual architecture, combined with the Nested Training strategy, effectively addresses the inefficiency of a single large MLP. This design leads to notably faster training and inference times, making the method more practical for high-detail modeling. The paper includes clear experiments and detailed ablation studies that thoroughly justify the effectiveness of the proposed components. Crucially, the authors include additional experiments—such as surface reconstruction from posed images, training a textured SDF from images/noisy point clouds, and attribute mapping (e.g., neural texture, depth mapping)—which effectively demonstrate the method's versatility and practical utility.

**Weaknesses:**

The core idea of modeling shape using a coarse network plus a finer residual correction is very similar to existing displacement field methods (IDF). Although the authors mention a simplification (L129, avoiding interpolation), the novelty of the multi-scale residual architecture should be more clearly articulated and distinguished from similar prior work in the related work and discussion sections.

The quantitative evaluation is too limited, currently relying only on Table 1 across 4 meshes. A more comprehensive assessment is necessary using standard, more complex datasets, such as models from the Thingi10K dataset as in NGLOD, to fully validate the method's scalability and robustness in capturing complex geometry.

While the authors claim M-plicits enables fast inference through real-time multiscale sphere tracing and mesh extraction, the reported efficiency gains are not clearly superior when compared against recent works that combine Instant-NGP/hash-grids with Neus to modeling complex geometry with high efficiency. This lack of a distinct advantage should be discussed.

Minor: some related works are missing it would be helpful to cite and discuss the following two papers that also explore multi-scale implicit representations for 3D shapes:
[1] Dou, Y., Zheng, Z., Jin, Q., & Ni, B. (2023). Multiplicative Fourier Level of Detail. Proceedings of the IEEE/CVF Conference on Computer Vision and Pattern Recognition (CVPR).
[2] Saragadam, V., Tan, J., Balakrishnan, G., Baraniuk, R., & Veeraraghavan, A. (2022). MINER: Multiscale Implicit Neural Representations. European Conf. Computer Vision (ECCV).

**Questions:**

In L306, it is mentioned that different $w_0$ values are used in SIREN to capture specific frequency bands. Why is the choice of $w_0$ for the fine-level network dependent on the complexity of the shape? What would be the effect if a consistently high value for $w_0$ at the fine level?

The results in Table 1 show a counter-intuitive outcome for the Armadillo dataset, where the final, corrected fine shape yields a higher (worse) Chamfer Distance than the initial coarse shape alone. What is the underlying cause for this degradation in performance following the fine-level residual?

Sphere Tracing Generalization: The paper proposes a real-time multiscale sphere tracing method. Can this efficient tracing method be readily applied to accelerate inference in other existing multiscale INRs, or is it tightly coupled with the specific architecture and Nested Training scheme proposed in this paper?

---

> ### Author Response · Authors · 2025-11-27
>
> We sincerely thank the reviewer for the positive assessment of our submission. We are particularly encouraged by the reviewer’s recognition that the proposed multi-scale residual design substantially improves both training and inference efficiency by avoiding reliance on a single high-capacity MLP, a central motivation of our work. Below we address each concern in detail.
>
> > The core idea of modeling shape using a coarse network plus a finer residual correction is very similar to existing displacement field methods (IDF). Although the authors mention a simplification (L129, avoiding interpolation), the novelty of the multi-scale residual architecture should be more clearly articulated and distinguished from similar prior work in the related work and discussion sections.
>
> IDF uses a base SDF and a single **displacement field** (both parameterized by SIRENs), whose composition with the base SDF produces a refined SDF. Such INR modeling is not efficient because the composition of MLPs requires sequential operations.
>
> On the other hand, M-plicits uses a residual sum of SIRENs (enabling more than two levels of detail), which allows parallel inference and thus real-time rendering. The key idea of M-plicits is the nesting scheme for training this residual sum to form a multi-scale SDF representation.
>
> We also present several components that enable the real-time application of our surface representation. Those components include a multiscale sphere tracing algorithm, GEMM-based normal computation, and the nesting condition that enables them. In other words, IDF cannot run multiscale ST safely (no nesting condition), provide smooth normals via analytic GEMM differentiation, or accelerate marching cubes via SDF culling. All real-time aspects of M-plicits depend on the SDF nesting and the normal-aligned training region — **not** on having a residual.
>
> It is also important to note, as discussed in the first paragraph of Sec. 4.2, that IDF’s residual baseline uses significantly larger networks (4 hidden layers × 256 neurons for each residual), whereas our networks are much smaller (coarse: 1 hidden layer × 128 neurons, medium: 1 hidden layer × 256 neurons, fine: 1 hidden layer with 400 neurons). For fairness, we tested our architecture with the IDF loss and sampling (last column), showing that even with smaller networks, our architectural choices are comparable in reconstruction quality on the Thingi32 dataset. Finally, when using our full scheme (architecture + loss + sampling), we achieve consistently strong results. This demonstrates that the nested-neighborhood supervision strategy, combined with our tailored loss, is a key factor in our performance gains. The table below shows the Chamfer Distance for a subset of the Thingi32 dataset.
>
> | Samples | CD (IDF arch & loss) | CD (ours arch + IDF loss) | CD (full M-plicits) |
> |--------:|-----------------------:|----------------------------:|----------------------:|
> | 44234 | 6.60E-02 | 7.20E-02 | **4.60E-04** |
> | 64764 | 6.10E-02 | 7.50E-02 | **2.10E-02** |
> | 68381 | 3.80E-02 | 4.10E-02 | **8.40E-03** |
> | 72870 | 6.30E-02 | 3.60E-02 | **2.20E-03** |
> | 73075 | 9.00E-02 | 9.20E-02 | **6.20E-02** |
> | 77245 | 7.30E-02 | 9.20E-02 | **3.60E-02** |
> | 354371| 3.80E-02 | 3.10E-02 | **1.70E-03** |

---

> ### Author Response · Authors · 2025-11-27
>
> > The quantitative evaluation is too limited, currently relying only on Table 1 across 4 meshes. A more comprehensive assessment is necessary using standard, more complex datasets, such as models from the Thingi10K dataset as in NGLOD, to fully validate the method's scalability and robustness in capturing complex geometry.
>
> > While the authors claim M-plicits enables fast inference through real-time multiscale sphere tracing and mesh extraction, the reported efficiency gains are not clearly superior when compared against recent works that combine Instant-NGP/hash-grids with Neus to modeling complex geometry with high efficiency. This lack of a distinct advantage should be discussed.
>
> > The results in Table 1 show a counter-intuitive outcome for the Armadillo dataset, where the final, corrected fine shape yields a higher (worse) Chamfer Distance than the initial coarse shape alone. What is the underlying cause for this degradation in performance following the fine-level residual?
>
> We evaluated M-plicits on Thing32, a subset of Thing10k commonly used by the community. Table 5 and the first paragraph of the ablation section present a comparison under a homogeneous setting, showing that even with fixed network sizes, our representation yields strong results.
>
> We further provide additional experiments, including IoU metrics on the Stanford dataset. Except for the Armadillo, all other models provide better metrics for the fine cases. It is also important to note that other methods may also suffer from worse metrics for the fine cases. However, the qualitative experiments in Figure 16 (revised manuscript) show that the fine level of our M-plicits (Multiscale ST column) is perceptually superior to the coarse level. For example, Instant-NGP presents similar values between the coarse and fine models for the Armadillo and performs worse for the Lucy model.
>
> | Model | IoU (⬆) |
> |--------------------------------|------------:|
> | Armadillo (ours coarse) | **2.60E-01** |
> | Armadillo (ours fine) | 1.91E-01 |
> | Armadillo (iNGP coarse) | 2.34E-02 |
> | Armadillo (iNGP fine) | 2.33E-02 |
> | Asian Dragon (ours coarse) | 7.14E-01 |
> | Asian Dragon (ours fine) | **8.81E-01** |
> | Asian Dragon (iNGP coarse) | 8.47E-03 |
> | Asian Dragon (iNGP fine) | 7.87E-03 |
> | Lucy (ours coarse) | 3.46E-01 |
> | Lucy (ours fine) | **6.63E-01** |
> | Lucy (iNGP coarse) | 1.96E-03 |
> | Lucy (iNGP fine) | 2.07E-03 |
> | Thai Statue (ours coarse) | 7.54E-02 |
> | Thai Statue (ours fine) | **9.20E-02** |
> | Thai Statue (iNGP coarse) | 1.17E-02 |
> | Thai Statue (iNGP fine) | 1.26E-02 |
>
>
> We also conducted an ablation on the number of residual levels, increasing the hierarchy to six levels. The table below reports the Chamfer distance for each level on the Buddha model (Stanford dataset) and a subset of Thing10k. In all experiments, Chamfer distance decreases monotonically up to Level 3, which justify our design choice. It is important to note that our approach does not claim to be a level-of-detail method, but a representation with fixed different scales that enable fast and accurate distance and attribute (normals and texture color) queries.
>
>
> | Model | Lvl 1 CD | Lvl 2 CD | Lvl 3 CD | Lvl 4 CD | Lvl 5 CD | Lvl 6 CD |
> |---------|----------:|----------:|----------:|----------:|----------:|----------:|
> | Buddha | 2.90E-03 | 2.29E-03 | **2.27E-03** | 2.29E-03 | 3.32E-03 | 2.17E-02 |
> | 44234 | 1.41E-03 | 1.08E-03 | **1.06E-03** | 4.65E-02 | 6.06E-02 | 6.18E-02 |
> | 64764 | 2.13E-02 | 2.10E-02 | **2.08E-02** | 4.88E-02 | 3.77E-02 | 3.53E-02 |
> | 68381 | 8.77E-03 | 8.58E-03 | **8.46E-03** | 3.17E-02 | 3.48E-02 | 3.58E-02 |
> | 72870 | 2.83E-03 | 2.06E-03 | **1.85E-03** | 3.25E-02 | 4.61E-02 | 5.18E-02 |
> | 73075 | 1.43E-03 | 1.22E-03 | **1.07E-03** | 1.81E-02 | 6.93E-02 | 9.59E-02 |
> | 77245 | 3.62E-02 | 3.57E-02 | **3.54E-02** | 7.83E-02 | 5.47E-02 | 5.13E-02 |
> | 354371 | 3.25E-03 | 2.80E-03 | **2.79E-03** | 3.98E-03 | 3.68E-02 | 2.94E-02 |
>
>
> We additionally evaluated our method on a *scene-scale point cloud* containing more than 10 million points. Figure 16 in the revised paper shows that our approach is able to reconstruct the scene in all proposed scales (coarse, medium, and fine) while Instant-NGP fails under both configurations tested (3 levels and 16 levels). We also present metrics (Chamfer, Hausdorff, and IoU) that corroborate those results in the following table. We achieve 2 or 3 orders of magnitude better Chamfer distance and one order of magnitude better IoU.
>
>
> | Method | Chamfer (⬇) | Hausdorff (⬇) | IoU (⬆) |
> |-------------------|-------------:|---------------:|-----------:|
> | Ours (coarse) | 5.02E-05 | **1.11E-02** | 2.46E-01 |
> | Ours (medium) | 4.48E-05 | 1.14E-02 | 2.69E-01 |
> | Ours (fine) | **3.87E-05** | **1.11E-02** | **3.83E-01** |
> | iNGP (3 levels) | 6.93E-03 | 5.98E-02 | 3.05E-02 |
> | iNGP (16 levels) | 1.12E-02 | 7.57E-02 | 2.44E-02 |

---

> ### Author Response · Authors · 2025-11-27
>
> We also evaluated robustness to noise. In this experiment, each vertex is randomly perturbed in the direction of the normal by up to 1.0% of the model bounding box diagonal.  Figure 10 (revised manuscript) shows the results for the Lucy model. The coarse level of M-plicits naturally acts as a **low-pass filter**, removing most of the high-frequency noise and providing a clean prior for subsequent levels. Instant-NGP (coarse/fine) fails to denoise the input and produces highly irregular surfaces. Quantitative results for three models (Chamfer, Hausdorff, IoU) further confirm the robustness of our approach (see Table 4).
>
> | Model | Chamfer (⬇) | Hausdorff (⬇) | IoU (⬆) |
> |-------------------------------|-------------:|---------------:|-----------:|
> | Armadillo (ours coarse) | 1.83E-03 | 1.18E-02 | 5.68E-02 |
> | Armadillo (ours medium) | **5.86E-05** | **7.18E-04** | **4.89E-01** |
> | Armadillo (ours fine) | 2.37E-03 | 1.21E-02 | 5.09E-02 |
> | Armadillo (iNGP coarse) | 1.75E-02 | 1.05E-01 | 3.54E-02 |
> | Armadillo (iNGP fine) | 1.93E-02 | 9.49E-02 | 3.34E-02 |
> | Asian Dragon (ours coarse) | 3.49E-04 | 1.55E-02 | 2.97E-01 |
> | Asian Dragon (ours medium) | 2.02E-04 | 4.10E-03 | 3.70E-01 |
> | Asian Dragon (ours fine) | **1.69E-04** | **2.76E-03** | **4.00E-01** |
> | Asian Dragon (iNGP coarse) | 2.35E-02 | 2.21E-01 | 4.70E-03 |
> | Asian Dragon (iNGP fine) | 1.62E-02 | 1.52E-01 | 6.68E-03 |
> | Lucy (ours coarse) | 3.33E-04 | 2.21E-02 | 3.02E-01 |
> | Lucy (ours medium) | 4.39E-05 | **8.31E-04** | 5.18E-01 |
> | Lucy (ours fine) | **3.45E-05** | 8.54E-04 | **5.98E-01** |
> | Lucy (iNGP coarse) | 2.36E-02 | 1.36E-01 | 0.00E+00 |
> | Lucy (iNGP fine) | 1.12E-02 | 7.57E-02 | 1.00E-03 |
>
> > Minor: some related works are missing it would be helpful to cite and discuss the following two papers that also explore multi-scale implicit representations for 3D shapes: [1] Dou, Y., Zheng, Z., Jin, Q., & Ni, B. (2023). Multiplicative Fourier Level of Detail. Proceedings of the IEEE/CVF Conference on Computer Vision and Pattern Recognition (CVPR). [2] Saragadam, V., Tan, J., Balakrishnan, G., Baraniuk, R., & Veeraraghavan, A. (2022). MINER: Multiscale Implicit Neural Representations. European Conf. Computer Vision (ECCV).
>
> Thank you for pointing out that additional context. We added and contextualized the references in the Multiscale Neural SDFs paragraph in the Related Works section of the revised paper.
>
> > In L306, it is mentioned that different $\omega_0$ values are used in SIREN to capture specific frequency bands. Why is the choice of $\omega_0$ for the fine-level network dependent on the complexity of the shape? What would be the effect if a consistently high value for $\omega_0$ at the fine level?
>
> For SIREN networks, $\omega_0$ controls frequency scale and sensitivity since it scales the pre-activation signal before the sine activation is applied. Specifically, SIRENs trained with lower $\omega_0$ values capture lower frequencies while SIREN with higher $\omega_0$ values capture higher frequencies. This is the intuition behind our choice of smaller values for the coarse network and higher values for the fine network.
>
> > Sphere Tracing Generalization: The paper proposes a real-time multiscale sphere tracing method. Can this efficient tracing method be readily applied to accelerate inference in other existing multiscale INRs, or is it tightly coupled with the specific architecture and Nested Training scheme proposed in this paper?
>
> Yes, it is generalizable for other representations if the nesting condition is enforced. Otherwise, false negatives may occur. In other words, a ray that should intersect a finer scale may miss if the coarse scale does not intercept it.

---

### Author Response · Authors · 2025-11-29
**Final Remarks to AC**

We thank the reviewers and the Area Chair for their thoughtful and constructive evaluations. We have addressed **all questions and concerns** raised during the review process, clarifying that the key conceptual contribution of **M-plicits** is not the use of multiscale residuals alone, but their combination with a novel **nested neighborhood training**, which guarantees SDF nesting across levels. This design enables strong compact SDF reconstruction while providing capabilities that previous methods do not simultaneously offer: real-time multiscale sphere tracing, fast analytic normals via a single forward pass, accelerated marching cubes through SDF culling, and texture support. We contextualize our method with IDF, BANF, NGLOD, and Instant-NGP, highlighting differences in training (nested vs. cascaded), representation (MLP-based vs. grid-based), and robustness.

To address concerns regarding evaluation, we expanded our experiments to include:

1. results on a sample of **Thing32** under a homogeneous setting;

2. **IoU metrics** on the Stanford dataset;

3. a **6-level ablation** confirming that CD improvements saturate at 3 levels;

4. a **scene-scale point cloud** (>10M points) where M-plicits outperforms Instant-NGP in CD and IoU; and

5. **robustness-to-noise** experiments, where M-plicits acts as a natural low-pass filter and remains stable, while Instant-NGP degrades significantly.

We also provide an ablation on the ε neighborhood parameter and clarify implementation details (network sizes, the number of samples per level, and the corrected medium-level width).

---

### Meta-Review · Area_Chair_YRNx · 2026-01-06

**Summary:**

The primary concerns leading to the final decision to reject center on the perceived limited conceptual novelty of the proposed method and insufficient empirical evidence to substantiate its claimed advantages. Reviewers consistently noted that the core components—multiscale residual architectures and nested or cascaded training—are incremental extensions of established ideas in prior works like IDF, BANF, and NGLOD. While the rebuttal provided extensive new experiments, including results on larger datasets and robustness tests, they did not convincingly demonstrate a qualitative leap in reconstruction accuracy over strong, efficient baselines like Instant-NGP, particularly for the fine-scale reconstruction as noted in the Armadillo results. The improvements, while consistent, were often marginal in key metrics, undermining the argument for a significant technical advance.

**Reviewer Concerns:**

The rebuttal effectively addressed several specific empirical concerns by adding evaluations on the Thingi32 dataset, IoU metrics, a 6-level ablation, and scene-scale reconstruction. Questions about parameter sensitivity and the number of training points were also clarified. However, the most critical outstanding concerns are twofold. First, the perceived novelty remains in question; the authors' defense that their contribution lies in the specific combination of residual sums with nested training, enabling new capabilities like safe sphere tracing, did not fully alleviate reviewer skepticism about the fundamental conceptual advance. Second, the practical advantage over modern efficient baselines (e.g., Instant-NGP) in terms of reconstruction quality was not decisively proven, with some results showing negligible or even negative gains at the finest level.

**Reviewer Scores:**

They did not respond during the rebuttal period.

---

### Decision · Program_Chairs · 2026-01-26

Reject